# Autoformulation of Mathematical Optimization Models Using LLMs

Nicolás Astorga [* 1]   Tennison Liu [* 1]   Yuanzhang Xiao [2]   Mihaela van der Schaar [1]

## Abstract

Mathematical optimization is fundamental to decision-making across diverse domains, from operations research to healthcare. Yet, translating real-world problems into optimization models remains a difficult task, often demanding specialized expertise. This paper approaches the problem of *autoformulation*: the automated creation of solver-ready optimization models from natural language problem descriptions. We identify three core challenges of autoformulation: *(1)* the vast, problem-dependent hypothesis space, *(2)* efficient and diverse exploration of this space under uncertainty, and *(3)* evaluation of formulation correctness against problem description. To address these challenges, we present a novel method leveraging *Large Language Models* (LLMs) with *Monte-Carlo Tree Search*, exploiting the hierarchical nature of optimization modeling to generate and systematically explore possible formulations. To enhance search efficiency, we introduce symbolic pruning to eliminate trivially equivalent search paths (branches), and employ LLM-based evaluation of partial formulations to guide search. Empirical analysis on linear and mixed-integer programming benchmarks demonstrates our method's effectiveness, with significant performance gains from both LLM-based value estimation and symbolic pruning techniques.

## 1. Introduction

Mathematical optimization has long been a cornerstone of decision-making processes across various domains, from supply chain management (Bramel & Simchi-Levi, 1997) and healthcare resource allocation (Delgado et al., 2022) to portfolio optimization (Mokhtar et al., 2014). These prob-

lems are characterized by maximizing an objective function subject to constraints (Williams, 2013). Traditionally, optimization modeling follows a three-step process: ▶ gathering problem requirements, typically expressed in unstructured formats and domain terminology; ▶ formulating these requirements into a formal mathematical model, including variables, constraints, and objective functions; ▶ implementing the model computationally using specialized modeling language for solution using commercial solvers.

**Autoformulation.** Despite major advances in solving algorithms over the past decades, the process of formulating optimization models still relies largely on human expertise to understand problem requirements and translate them into mathematical programs that software can efficiently solve to find optimal decision values. Autoformulation aims to address this bottleneck by automating the formulation process, with the potential to significantly improve the time- and cost-efficiency of the modeling process. For *modelers*, autoformulation assists with rapid prototyping and iteration of different formulations, reducing development time and costs while minimizing implementation errors. For *domain experts*, it makes optimization tools accessible without requiring deep optimization expertise, allowing domain experts to focus on critical business aspects like requirements gathering, use-case development, and communication.

At its core, we conceptualize autoformulation as a search for an optimal formulation within a vast hypothesis space. This search faces several key challenges. First, the hypothesis space is large and problem-dependent, encompassing diverse variable definitions, constraint structures, or objective function forms, with complex dependencies between these modeling decisions. Second, efficiently navigating this space requires balancing exploitation and exploration, particularly given the uncertainty in correct formulations and redundancy in the hypothesis space. Finally, like any search process, autoformulation requires a reliable evaluation mechanism for candidate solutions. While solvers can assess optimality and computational efficiency, determining whether a formulation accurately captures the intended real-world problem remains particularly challenging.

Recent works (Ramamonjison et al., 2023; Xiao et al., 2023; AhmadiTeshnizi et al., 2024) have demonstrated the promising potential of *Large Language Models* (LLMs) in autofor-

---
[*]Equal contribution  [1]DAMTP, University of Cambridge, Cambridge, UK  [2]ECE, University of Hawaii at Manoa, Honolulu, USA. Correspondence to: Nicolás Astorga, Tennison Liu <{nja46,tl522}@cam.ac.uk>.

*Proceedings of the 42nd International Conference on Machine Learning*, Vancouver, Canada. PMLR 267, 2025. Copyright 2025 by the author(s).

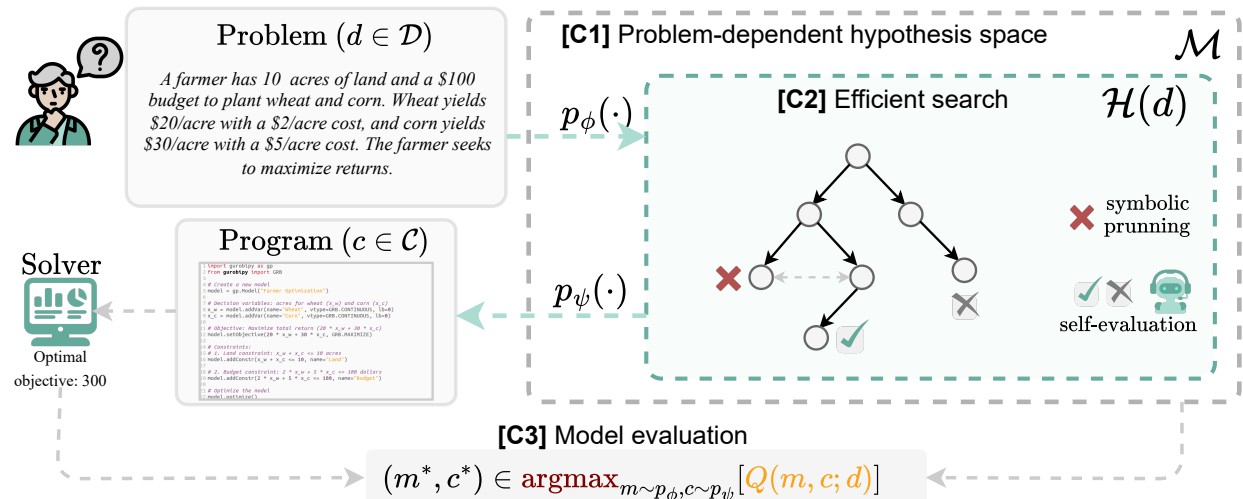

*Figure 1.* **Autoformulation and its challenges.** Autoformulation translates a problem description ($d \in \mathcal{D}$) into mathematical ($m \in \mathcal{M}$) and computational ($c \in \mathcal{C}$) models. The challenges include **[C1]** vast, problem-dependent hypothesis space, **[C2]** efficient search under formulation uncertainty and redundancy, and **[C3]** evaluating formulation correctness against problem requirements.

mulation, laying important groundwork in this field. LLMs contribute several crucial capabilities to this process: natural language understanding of problem descriptions, vast domain knowledge to incorporate relevant modeling techniques, and in-context learning and reasoning capabilities (Brown, 2020; Chowdhery et al., 2023). Building upon these contributions, our work focuses on developing techniques for efficient, systematic exploration and introducing robust mechanisms for evaluating formulation correctness.

**Key considerations.** By conceptualizing autoformulation as a search problem, we exploit the inherent hierarchical structure of optimization modeling to efficiently search through the problem-dependent hypothesis space, guided by feedback on formulation correctness. Our first innovation is to **(1)** decompose optimization modeling into hierarchical components and develop a *Monte-Carlo Tree Search* (MCTS) method to incrementally explore each component's formulation space (Coulom, 2006). This exploration is powered by LLMs serving as conditional hypothesis generators, creating diverse component formulations at each level of the search. To improve search performance, we introduce two additional innovations: **(2)** a pruning technique using *Satisfiability Modulo Theories* solvers (Barrett & Tinelli, 2018), to eliminate redundant hypotheses (i.e., syntactically different yet functionally equivalent); and **(3)** LLM-based evaluators of formulation correctness, combined with solver feedback, to obtain a reward signal to guide efficient search.

**Contributions.** Our main contributions are: ① We formalize *autoformulation* of mathematical optimization models as a search problem and identify its core challenges. ② We develop a novel approach combining LLMs, symbolic tools with MCTS to enable efficient and systematic exploration of the optimization model space, using LLMs as both hypoth-

esis generators and correctness evaluators alongside symbolic pruning techniques. ③ We demonstrate our method's superior performance across two real-world benchmarks containing linear and mixed-integer programming problems, observing significant performance gains from both search pruning and LLM-based formulation evaluation.

## 2. Autoformulation: Towards Automated Optimization Modeling

Optimization modeling seeks to minimize an objective function subject to specific constraints on decision variables (Dantzig, 1990). The mathematical model can be expressed in a **general form**:

$$
\begin{aligned}
\text{Minimize} \quad & f(\mathbf{x}) \\
\text{subject to} \quad & g_i(\mathbf{x}) \leq 0, \quad i = 1, \ldots, I, \\
& h_j(\mathbf{x}) = 0, \quad j = 1, \ldots, J.
\end{aligned}
\tag{1}
$$

Here $\mathbf{x} \in \mathcal{X}$ represents the vector of decision variables, and $\mathcal{X} \subseteq \mathbb{R}^\ell \times \mathbb{Z}^k$ is the domain for which the objective and constraints functions are all defined. Furthermore, $f : \mathcal{X} \to \mathbb{R}$ is the objective function to be minimized, $g_i : \mathcal{X} \to \mathbb{R}$ are inequality constraints, $h_j : \mathcal{X} \to \mathbb{R}$ are equality constraints, and $I$ and $J$ are the numbers of inequality and equality constraints respectively. The feasible region is the set of all possible points that satisfy the problem constraints: $\{\mathbf{x} \in \mathcal{X} \mid g_i(\mathbf{x}) \leq 0, \ \forall i \in [I], h_j(\mathbf{x}) = 0, \ \forall j \in [J]\}$.

**Convex problems.** An optimization problem is **convex** if $f$ and $g_i \ \forall i \in [I]$ are convex, and $h_j \ \forall j \in [J]$ are affine. Convexity is significant as any local optimum of a convex problem is globally optimal, and specialized solvers can efficiently solve convex problems to global optimality using advanced algorithms (e.g., Gurobi (Gurobi Optimization,

LLC, 2024), CVXPY (Diamond & Boyd, 2016)). Before utilizing these solvers, the mathematical models are first represented in code as computational models, which are then passed to the solvers for optimization.

## 2.1. Problem Definition

Boyd & Vandenberghe (2004) aptly recognized that *"the challenge, and art, in using convex optimization is in recognizing and formulating the problem. Once this formulation is done, solving the problem is ... (almost) technology"*. While solver technology has significantly matured, the process of formulating optimization models remains largely human expertise driven. Responding to this challenge, *autoformulation* is the automated process of transforming natural language descriptions of real-world problems into formal optimization models, thus automating the "challenge and art" of problem formulation.

---

**Autoformulation: Formal Definition**

Let $\mathcal{D}$, $\mathcal{M}$, and $\mathcal{C}$ represent the spaces of natural language problem descriptions, mathematical formulations, and computational models respectively, with $d \in \mathcal{D}$, $m \in \mathcal{M}$, and $c \in \mathcal{C}$ as their elements. Autoformulation involves two transformations:

1. **Mathematical formulation** $p_\phi : \mathcal{D} \to P(\mathcal{M})$: Transforming problem description into a mathematical formulation. Here, $P(\cdot)$ represents the space of probability distributions.
2. **Computational representation** $p_\psi : \mathcal{M} \to P(\mathcal{C})$: Converting the mathematical formulation into computational formats suitable for solvers, which includes representing the model in a programming framework, and specifying a solving algorithm.

**Autoformulator.** Here, $p_\phi$ and $p_\psi$ are models of each transformation, with $\phi$, $\psi$ their respective parameters.[a] The complete autoformulation process can thus be represented as inferring the joint distribution $p_{\phi,\psi}(m, c \mid d) = p_\psi(c \mid m) \cdot p_\phi(m \mid d)$. We refer to any algorithm designed for autoformulation as an *autoformulator*.

**Objective.** For a given problem $d$, the autoformulator aims to find optimal mathematical and computational formulations that maximize an evaluation measure $Q(\cdot)$ :

$$(m^*, c^*) \in \underset{m \sim p_\phi, c \sim p_\psi}{\arg\max} \ Q(m, c; d) \qquad (2)$$

**Evaluation criteria.** Here, $Q$ assesses the quality of $(m, c)$ relative to $d$. There are many possible instantiations of $Q$, a primary example is **formulation correctness**—accuracy of the formulation in reflecting problem requirements. Given that a formulation is correct, other measures could consider **optimality gap** (distance from optimal value, where certain convex formulation can

---

achieve zero optimality gap), and **computational efficiency** (solution time and resource requirements, which can vary significantly between equivalent formulations).[b]

---

[a]Following convention (Sumers et al., 2024), we define the weights and procedural prompts as the *parameters ($\phi$, $\psi$)* of an LLM-based autoformulator.

[b]In Appendix E, we discuss and empirically analyze the effects of problem (re)-formulation and solver configuration on optimality and computational efficiency.

---

**A few observations.** While autoformulation involves two transformations, the mathematical formulation step ($p_\phi$) generally presents significantly greater challenges than creating computational models ($p_\psi$). This has also been observed empirically in recent studies, where formulating mathematical models was the primary source of errors (Xiao et al., 2023; AhmadiTeshnizi et al., 2024). Indeed, this step requires deep domain understanding, abstraction of real-world complexities into mathematical constructs, and a certain creativity in effective reformulations. While the translation to computational models often follows a more standardized pattern, with some automation already available through commercial packages (Fourer et al., 1990). Thus, our subsequent analysis focuses primarily on the mathematical formulation step. However, we note that the second transformation also presents unique challenges, most notably through the choice of solving algorithm and its hyperparameter configuration.

## 2.2. Challenges

Our conceptualization of autoformulation as a search problem reveals a few key challenges:

**[C1] Problem-dependent hypothesis space:** For each problem $d$, there exists a vast and problem-dependent hypothesis space $\mathcal{H}(d)$, encompassing various variable definitions, constraint structures, objective functions, and their interdependencies. This interdependence and domain-specificity makes it infeasible to manually enumerate or construct the search space (required in traditional search problems), requiring automated methods to generate valid and interdependent modeling components.

**[C2] Efficient search:** Efficiently searching the hypothesis space is challenging, as 'good' formulations can be sparse. There are two key **uncertainties** complicating this search: uncertainty in formulation choices and uncertainty due to ambiguous requirements (e.g., implicit or common-sense constraints, including non-negativity of resource constraints). Additionally, **trivial model equivalence**—syntactically different but functionally identical formulations (e.g., $2x + 3y$ versus $3y + 2x$)—can lead to inefficient exploration of superficial variations at the expense of discovering more diverse and valuable formulations. Here, 'trivial' refers to syntactic variations, distinct from mathematical reformulations that change the underlying structure

(e.g., converting non-convex to convex constraints).

**[C3] Model evaluation:** While solvers can assess computational aspects like efficiency and solvability, evaluating **formulation correctness**, whether a model faithfully captures the intended problem requirements, remains a core challenge. This absence of a correctness signal complicates the search process, as an efficient and optimal solution to an incorrectly formulated problem is ultimately invalid.

Here, we note that the complexity of autoformulation also varies significantly with problem characteristics of $d$, particularly convexity properties—while some problems allow direct solution for global optimality, others require the autoformulator to identify convex reformulations or develop relaxation strategies balancing optimality and computational efficiency (please see Appendix C for a detailed discussion).

# 3. LLM-Enhanced MCTS Search for Autoformulation

**Overview.** Recent developments have shown the promising potential of using *Large Language Models* (LLM) for autoformulation, leveraging their ability to generate formulations dynamically and bypassing the need of manually constructing hypothesis spaces (**[C1]**, Xiao et al. (2023); AhmadiTeshnizi et al. (2024)). Our approach builds on these works, differing in three key ways. First, we decompose the search space using optimization modeling's hierarchical structure, enabling systematic exploration through *Monte-Carlo Tree Search* (MCTS) rather than generating complete formulations at once **[C2]**. Second, we enhance efficiency by combining LLM-based evaluation of partial and complete formulations with symbolic pruning of equivalent branches, reducing redundant exploration while improving search guidance **[C3]**. Third, we employ a deterministic parser to automatically transform mathematical models into solver-ready computational code. While this successfully handled all problems in our experiments, eliminating a source of error where LLM-based translation (as used in existing works) proved unnecessary, we acknowledge that more complex transformations may require sophisticated approaches in future work. In what follows, we discuss each aspect of our method in turn.

## 3.1. Hierarchical Decomposition

Optimization modeling is inherently complex, involving multiple interconnected components. To manage this complexity and improve search efficiency, we propose a decomposition of the formulation process. This approach allows us to sequentially explore each model component rather than searching for entire formulations at once, potentially leading to more efficient search.

Specifically, we structurally decompose the autoformula-

tion process into four distinct stages, each represented by $m_i$. The complete mathematical formulation is defined as $m = \oplus_i^4 m_i$, where $\oplus$ denotes the composition of model components: $m_1$—**parameters and decision variables**, $m_2$—**objective function**, $m_3$—**equality constraints**, and $m_4$—**inequality constraints**. Given a problem $d$, the joint distribution $p_{\phi,\psi}(c, m \mid d)$ is decomposed hierarchically:

$$p_{\phi,\psi}(c, m \mid d) = p_\psi(c \mid m) \prod_{i=1}^{4} p_\phi(m_i \mid m_{<i}, d) \quad (3)$$

Here, $p_\phi(m_i \mid m_{<i}, d)$ represents the sequential nature of mathematical formulation, where each component $m_i$ depends on the *partial formulation* $m_{<i} = \oplus_{j=0}^{i-1} m_j$ (with $m_0 = \emptyset$) and the problem description $d$.

## 3.2. MCTS-Based Autoformulator

Having established a structured decomposition of the autoformulation process, we now address the challenge of efficiently navigating this hierarchical space. We employ an MCTS-based algorithm, which is particularly well-suited for exploring complex, hierarchical search spaces (Coulom, 2006). Our MCTS constructs a search tree of depth 4 to explore possible formulations, where each of the four levels corresponds to a component in our structured decomposition ($m_1$ to $m_4$). Nodes in this tree contain component formulations, and a complete formulation is represented by a path from the root to a terminal node.

The MCTS algorithm iteratively builds the search tree through four key steps: ▶ **selection**, ▶ **expansion**, ▶ **evaluation**, and ▶ **backpropagation**. For notational clarity, we denote a tree node as $n$ and any of its child nodes as $n_{child} \in Child(n)$, where $Child(n)$ is the set of all child nodes of $n$. We use $\vec{n}$ to represent the *partial* formulation by concatenating the path from root to node $n$. For instance, $\vec{n}$ for a node of depth 2 is the partial formulation containing the parameters, decision variables, and the objective function. Terminal nodes are denoted as $n_t$. In the interest of space, we present detailed information about all prompts used in the algorithm in Appendix B, providing only high-level details in the following subsections.

### 3.2.1. EXPANSION

Upon reaching an unexpanded node $n$, we generate its child nodes $Child(n)$ through expansion. Unlike traditional MCTS, which expands all actions from a predefined space, our expansion explores an *undefined* space of possible component formulations. We leverage LLMs to generate these potential formulations, using the partial formulation constructed so far as context to ensure coherent expansions. Our process involves: **(1)** generating diverse candidate formulations through LLM-based exploration, and **(2)** pruning trivially equivalent candidates to maintain a manageable yet

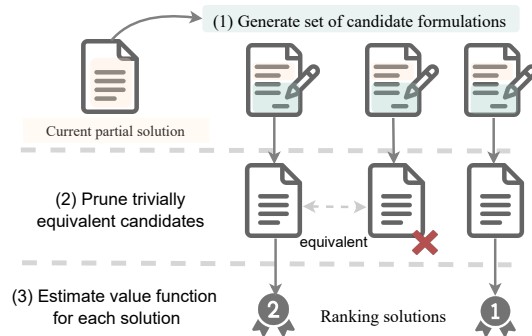

*Figure 2.* **Expansion and evaluation.** Expansion involves generating candidate formulations, which are then pruned to remove trivial equivalences. Remaining candidates are assigned a normalized rank score as value initialization.

diverse search space. The combined expansion and evaluation process is illustrated in Figure 2.

**Generating candidate formulation.** At node $n$, the LLM generates potential child nodes (containing formulations of the next component formulations) by conditioning on the partial formulation and problem description, which we denote as $\texttt{LLM}_\phi(n_{child} \mid \vec{n}, d)$. The LLM is queried through a structured prompt with three elements: ► problem description: the original natural language problem description $d$; ► partial formulation: the current partial formulation $\vec{n}$ in JSON format; ► level-specific instructions: guidelines for the current modeling stage, including output format and relevant considerations. We represent formulations and request formulations using JSON format, where keys are descriptive labels and values are mathematical expressions. For example, when generating possible inequality constraints, the LLM might return the formulation: {*"material_balance"*: $x_1 + x_2 \leq 100$, *"quality_requirement"*: $0.8x_1 + 0.6x_2 \geq 75$}. For each node expansion, we sample $H \in \mathbb{N}$ hypotheses from the LLM's distribution: $\widehat{Child}(n) = \{\tilde{n}_{child}^{(h)} \mid \tilde{n}_{child}^{(h)} \sim \texttt{LLM}_\phi(\cdot \mid \vec{n}, d), \forall h \in [H]\}$, where $\tilde{n}_{child}^{(h)}$ represents the $h$-th candidate component formulation.

**Search pruning.** After generating candidate formulations, we prune the search space to ensure diversity and efficiency. Specifically, we eliminate *trivially equivalent formulations*—expressions that differ only in syntax while remaining functionally identical (e.g., $2x + 3y$ versus $3y + 2x$). This pruning operation can be expressed as $Child(n) = \texttt{pruning}(\widehat{Child}(n))$. We employ *Satisfiability Modulo Theories* (SMT) solvers to detect equivalent formulations (Barrett & Tinelli, 2018). For components $m_2$-$m_4$ (objective functions, equality, and inequality constraints respectively), we represent each candidate formulation as a system of equations or inequalities. To compare two systems $S_1$ and $S_2$ over domain $\mathcal{X}$, we check the satisfiability of $\neg(\forall \mathbf{x} : (S_1(\mathbf{x}) \iff S_2(\mathbf{x})))$. Unsatisfiability proves

equivalence by showing no $\mathbf{x}$ exists where the systems differ, while satisfiability indicates distinct formulations. We apply this check pairwise across candidates in $\widehat{Child}(n)$, pruning trivially equivalent ones. Detailed SMT formulae are provided in Appendix B.

While SMT solvers effectively detect equivalent formulations, their decidability varies across problem types (Monniaux, 2016). Linear arithmetic over real and integer domains is generally decidable, but mixed-integer or non-linear functions may be undecidable depending on problem properties. When a solver cannot determine equivalence, we *conservatively* treat formulations as distinct, potentially exploring some redundant paths but avoiding premature pruning. While this is a heuristic approach that has scope for future improvements, our empirical analysis indicates that it yields significant efficiency gains through pruning (Section 5.3). Additionally, since SMT solvers require consistent variable domains $\mathcal{X}$, we apply them only to levels $m_2$-$m_4$ where nodes share decision variables. For level $m_1$, which defines decision variables, we query an LLM for pruning.

### 3.2.2. EVALUATION

After expansion, each newly created child nodes undergoes an initial evaluation to estimate its value, guiding subsequent exploration. While it is possible to use uniform priors, we employ LLMs to evaluate child nodes to provide informed value estimates, helping guide search toward promising formulations earlier. Specifically, the LLM is provided with all partial formulations of newly expanded child nodes, namely $\{\vec{n}_{child} \mid n_{child} \in Child(n)\}$, and instructed to assign a numerical rank (from 1 to $|Child(n)|$), based on its evaluation of formulation correctness, constraint feasibility and alignment with the original problem description). These ranks are then center-normalized to $[0, 1]$, with the middle rank centered at $0.5$. We denote this normalized score $s(\vec{n}_{child})$, which is used to initialize the child node's value $V_{\text{prior}}(n_{child}) \leftarrow s(\vec{n}_{child})$. Subsequently, we retain the top $I \in \mathbb{N}$ candidates, based on their normalized rank scores.

### 3.2.3. DUAL REWARDS AND BACKPROPAGATION

**Terminal rewards.** We continue expanding until a terminal node $n_t$ is reached, where $\vec{n}_t$ represents a complete formulation (from root to terminal node). We evaluate the complete formulation using a dual approach, combining assessments of both mathematical correctness and computational model's performance to obtain *reward* $r(\vec{n}_t)$:

$$r(\vec{n}_t) = \mathbb{I}(E_{\texttt{solver}}^c(\texttt{parser}(\vec{n}_t)) = 1) \cdot E_{\texttt{LLM}}^m(\vec{n}_t; d) \quad (4)$$

where $\mathbb{I}$ is the indicator function and $c = \texttt{parser}(\vec{n}_t)$ is our custom parser that converts each mathematical formulation into a computational representation. $E_{\texttt{LLM}}^m(\vec{n}_t; d)$ is the LLM's evaluation of the mathematical formulation's

correctness, assessing how well it captures the problem requirements and constraints in $d$. $E^c_{\texttt{solver}}(\texttt{parser}(\vec{n}_t))$ is the solver's binary feedback on whether the program was solved optimally. We note that this is an imperfect signal, as an incorrectly formulated model could be solved to optimality despite not faithfully representing the original problem, highlighting the importance of dual evaluation.

To evaluate formulation correctness for complete models, we employ a comparative evaluation approach rather than independent scoring. While LLMs could directly assign scores to each formulation, our empirical analysis showed mixed results with this approach—likely due to scoring inconsistencies when evaluating solutions in isolation. In comparison, relative comparisons yield more robust and consistent evaluations. However, the approach described in Section 3.2.2 is no longer practical (as each new solution would require re-ranking and re-computing reward for all previous formulations). Instead, we introduce a comparative method where each formulation is evaluated against a consistent set of baseline models $m_b$. The LLM outputs a score in $[0, 1]$, where values above 0.5 indicate preference for the candidate formulation over the baseline. Formally, we express this as $E^m_{\texttt{LLM}}(\vec{n}_t; d) \sim \texttt{LLM}(\cdot \mid \vec{n}_t, m_b; d)$. This approach ensures comparable rewards across all terminal nodes by maintaining a consistent reference point.

**Backpropagation.** Following the reward calculation, we backpropagate this value to update the statistics of all nodes along the trajectory. For each node in this path from root to terminal node, $n_t$, we apply the following updates: $V_{\text{bp}}(n) \leftarrow \frac{V_{\text{bp}}(n) \cdot N(n) + r(\vec{n}_t)}{N(n)+1}, \quad N(n) \leftarrow N(n) + 1, \quad \forall\, n \in \vec{n}_t$. Here, we increment the visit count $N(n)$ by 1 and update the value $V_{\text{bp}}(n)$ with a weighted average of its previous value and the new reward $r(\vec{n}_t)$. This backpropagation process ensures that the tree gradually accumulates more accurate estimates of node values. These updated statistics then inform the selection strategy in subsequent iterations. The node value used in selection is then $V(n) = \lambda \cdot V_{\text{prior}}(n) + (1 - \lambda)V_{\text{bp}}(n)$.

3.2.4. SELECTION

The selection step guides the search towards promising regions of the tree. Starting from the root, the algorithm recursively selects child nodes using the Upper Confidence Bound for Trees (UCT): $n^*_{child} = \arg\max_{n_{child} \in Child(n)} \left( V(n_{child}) + \omega \sqrt{\frac{\ln N(n)}{N(n_{child})}} \right)$ (Kocsis & Szepesvári, 2006). This process continues until reaching an unexpanded node. Here, $n^*_{child}$ is the selected child node, $V(n_{child})$ is its estimated value, $N(n)$ and $N(n_{child})$ are visit counts for the parent and child nodes respectively and $\omega$ is an exploration constant. This formula balances exploitation (first term, favoring high-value nodes) with exploration (second term, favoring less-visited nodes).

**Summary.** Our MCTS-based algorithm iterates through the aforementioned steps, progressively constructing and refining a tree of possible formulations. We execute this process for $T \in \mathbb{N}$ iterations, thoroughly exploring the space of potential models and identifying promising formulations. The final output is a set of $M \in \mathbb{N}$, $M \leq T$ *functionally distinct* optimization models (achieved through search pruning), where each model is defined by a unique path through the tree. Formally, we express the overall algorithm as: $\{(m^{(i)}, c^{(i)}, r^{(i)})\}_{i=1}^M = \texttt{MCTS}_{\texttt{LLM}}(d)$. The superscript $i$ indexes the functionally distinct formulation, and $r^{(i)}$ is the estimated value/reward of the corresponding terminal node.

# 4. Related Work

**Advances in LLMs.** Recent works have demonstrated the substantial potential of LLMs in solving complex reasoning tasks, including language understanding (Hendrycks et al., 2021), commonsense reasoning (Brown, 2020), logical reasoning (Wei et al., 2022; Yao et al., 2024), mathematical problem-solving (Lewkowycz et al., 2022), and coding tasks (Chen et al., 2021). Of particular relevance are studies employing LLMs in optimization and search tasks, such as Bayesian Optimization (Liu et al., 2024), prompt optimization (Guo et al., 2023), evolutionary optimization (Yang et al., 2024; Liu et al., 2025), and symbolic program refinement (Madaan et al., 2024). In contrast, our focus is on leveraging LLMs to bridge the gap between natural language description and formal optimization models.

**Autoformulation.** Early work by Ramamonjison et al. (2023) introduced the first autoformulation competition. The competition focused on linear programming problems, but required predicting formulations in specific formats (i.e., entity problem tagging), using *pre*-LLM era NLP models with limited generalization beyond given formats. Recent advances by Xiao et al. (2023) and AhmadiTeshnizi et al. (2024) employed multi-agent LLM frameworks, where multiple agents (e.g., coding and formulation agents) collaborate to generate and iteratively refine complete formulations. Our approach differs by decomposing the formulation into components, using MCTS for systematic exploration, and incorporating symbolic pruning and composite rewards to improve search efficiency. In parallel, Tang et al. (2024) developed the first LLM specifically finetuned for optimization modeling using a mixture of real and synthetic data.

**Planning.** Recent research have also explored the integration of LLMs with planning algorithms (Huang et al., 2024a), the most pertinent of which consider approaches that *generate* and *select* from multiple plans (Wei et al., 2022; Wang et al., 2023). Such approaches are particularly effective for complex tasks, where a single plan generated by LLM is likely to be suboptimal, thus requiring exploration. Yao et al. (2023) employed an LLM to generate multiple rea-

*Table 1.* **Benchmark comparison.** Formulation correctness results on four benchmarks containing LPs/MILPs.

| Method | NL4OPT | IndustryOR | MAMO (ComplexLP) | ComplexOR |
|---|---|---|---|---|
| *Finetuned methods* | | | | |
| ORLM$_{\text{Llama3-8B}}$ | 85.7% | 38.0% | 39.3% | 44.4% |
| *Methods based on* GPT4 | | | | |
| Standard | 47.3% | 28.0% | 24.6% | 9.5% |
| Reflexion | 53.0% | - | 36.0% | 19.1% |
| Chain-of-Experts | 64.2% | - | 40.2% | 38.1% |
| OptiMUS | 78.8% | - | - | 66.7% |
| Autoformulator ($N$=1) | 85.24% | 35.0% | 43.8% | 66.7% |
| Autoformulator ($N$=3) | 92.21% | 42.0% | 61.4% | 72.2% |
| Autoformulator (All) | 92.62% | 48.0% | 62.3% | 72.2% |

soning paths and self-evaluating choices to decide the next action. Hao et al. (2023); Zhao et al. (2024) employ LLMs as policy functions in MCTS framework, where potential actions are generated through LLM calls. Plans are evaluated either through grounded feedback from the environment or LLM self-evaluation, including the probability of 'good' actions (Hao et al., 2023), or a continuous score (Yuan et al., 2024). Compared with these search-guided methods, our work differs in hierarchical search, SMT-based pruning, and comparative/ranking based evaluation of correctness, innovations specifically tailored to autoformulation.

# 5. Experiments

We present our experimental evaluation across three key areas. First, we benchmark our autoformulator against baseline approaches on real-world problems (Section 5.1). We then analyze two critical components: our ranking and comparative evaluation methods for assessing formulation correctness (Section 5.2), and our search space pruning techniques for improved efficiency (Section 5.3). Section 5.4 concludes with insights on exploration diversity, performance across problem types, and failure modes.

**Benchmarks.** We evaluate our methods on four real-world benchmarks: **NLP4OPT** (Ramamonjison et al., 2023), a curated set of 244 linear programming problems (based on (Tang et al., 2024)); **IndustryOR** (Tang et al., 2024), consisting of 100 problems spanning linear, integer, and mixed-integer programming at various difficulty levels; **ComplexOR** (Xiao et al., 2023), with 37 real-world operations research problems from diverse domains; and **MAMO** (Huang et al., 2024b), using the more advanced ComplexLP subset, which includes 211 problems.

**Evaluations.** Following (Tang et al., 2024; AhmadiTeshnizi et al., 2024), we report accuracy as the proportion of problems where the discovered formulation yielded optimal objective values. All baselines and experiments use GPT4-0613 as the underlying LLM.

## 5.1. Benchmark Comparisons

**Baselines.** We compare against several methods: zero-shot prompting (Standard), the reasoning-augmented Reflexion (Shinn et al., 2023), and three specialized autoformulators: Chain-of-Experts (Xiao et al., 2023) and OptiMUS (AhmadiTeshnizi et al., 2024), both based on multi-agent frameworks, and ORLM (Tang et al., 2024), a Llama3-based model finetuned on a mix of real and synthetic optimization datasets.

**Analysis.** We configure our method with $H = 10$ candidate formulations, $I = 3$ children retained after pruning and scoring, and $T = 16$ total rollouts. Unlike prior approaches that return a single model, our MCTS-based search generates up to $T$ distinct formulations. Accordingly, we report Pass@$N$ metrics to capture performance across multiple candidates. Results in Table 1 show that our method matches baseline performance with just one rollout, illustrating the efficiency of hierarchical search decomposition. With three rollouts, we surpass all baselines, including the finetuned ORLM model. While additional rollouts further improve accuracy, gains taper off due to diminishing returns and increased computational cost.

Additional results are provided in Appendix A, including comparisons with two ablated variants of our method that underscore the importance of structured tree search. We also present Best-of-N comparisons against ORLM (our closest competitor), where we select the candidate with the highest estimated value, highlighting the benefits of exploration in our framework.

## 5.2. Formulation Correctness Evaluation

Next, we examine our formulation evaluation methods and their effects on search performance. Specifically, we analyze the estimated reward of complete formulations, and the estimated value of partial formulations.

**(1) Complete formulation reward.** To analyze our approach to evaluate complete formulations, we considered

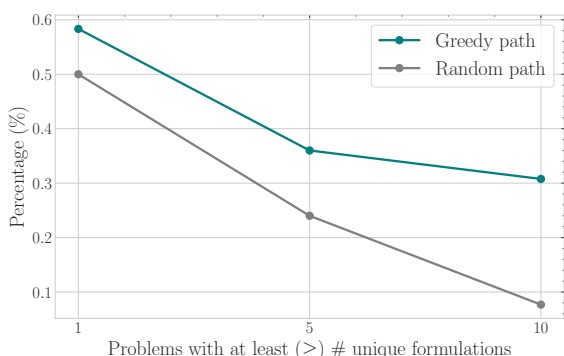

*Figure 3.* **Evaluation of value initialization.** Comparison of node selection based on initial value estimates *vs.* random selection.

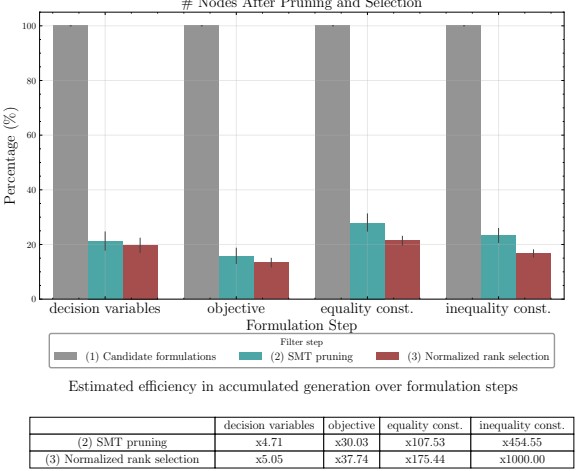

| | decision variables | objective | equality const. | inequality const. |
|---|---|---|---|---|
| (2) SMT pruning | x4.71 | x30.03 | x107.53 | x454.55 |
| (3) Normalized rank selection | x5.05 | x37.74 | x175.44 | x1000.00 |

*Figure 4.* **Improvements in search efficiency.** Pruning and selection by normalized rank score significantly reduces search space.

all problems in IndustryOR where a correct solution was found, and collected the scores assigned to correct and incorrect formulations. We found that correct solutions were evaluated with higher scores than incorrect formulations, obtaining a biserial correlation coefficient of $0.48$ ($p$-value of $2.0681e{-}3$). We compare this with a direct scoring method (Zhang et al., 2024) that independently scores each formulation from 1-100. This yielded a correlation of $0.23$ ($p$-value of $1.1185e{-}1$), underscoring the effectiveness of our comparative evaluation approach.

**(2) Partial formulation scores.** To evaluate estimated prior scores, we first obtain a fully expanded tree using Depth-First Search, where each node can have up to three children. Then we used our comparative evaluation to obtain node scores. For evaluation, we compared the correctness of a greedy formulation obtained by greedily selecting the highest scoring node in each level of the search tree and a randomly obtained solution. Figure 3 reveals that greedy solutions obtained from prior scores were significantly more accurate, with the gap increasing as the number of unique formulations contained in the tree increases.

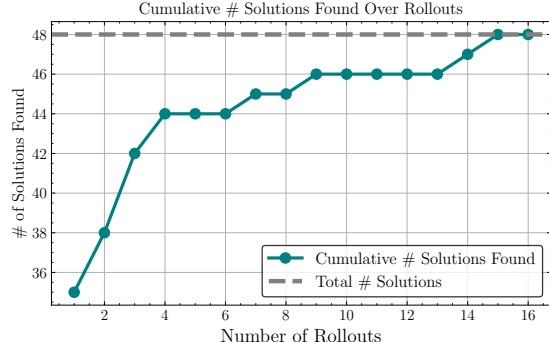

*Figure 5.* **Rate of formulation discovery.** Our method continues to discover unique and correct formulations during search.

### 5.3. Gains in Search Efficiency

The goal of this experiment is to analyze the gains in search efficiency. Recall that, in each formulation stage, candidate child nodes are **(1)** pruned using SMT to eliminate redundant formulations, and **(2)** selective retention of the top-3 candidates based on normalized rank scores. In Figure 4, we visualized the number of retained solutions of each filtering stage. We note that, on average, around $20\%$ of formulations are retained after the symbolic pruning stage, with further reduction after the selection stage. To quantify the efficiency gains, we compared our approach against a non-hierarchical search method. The analysis reveals that a non-hierarchical approach would require 1000x more formulation generations to produce the same number of unique formulations, demonstrating substantial savings in search budgets.

### 5.4. Performance Analysis

We conclude our evaluation by analyzing our method's performance through three lenses: **(1)** the rate of correct model discovery, **(2)** performance across problem categories, and **(3)** the underlying sources of error in formulated models.

**Formulation discovery rate.** Figure 5 describes the number of correct formulations discovered on IndustryOR as a function of MCTS rollouts (where $48$ total correct formulations were found). We observed that while additional rollouts consistently yielded more correct solutions, the search produces diminishing returns: discovering the final 4 correct solutions required 10 additional rollouts. This illustrates that while our method continues to explore useful candidates, the marginal gain per rollout decreases, highlighting a trade-off between coverage and computational budget.

**Finegrained performance.** We further examined accuracy across different problem categories in Table 2. Our method performs consistently well across categories, with no significant drop in performance for any specific type. Interestingly, we observed that categories with lower accuracy tended to exhibit greater average tree entropy, a measure of diversity in the generated search trees. This suggests that tree entropy

may be a useful indicator of uncertainty and a potential predictor of formulation success.

*Table 2.* **Finegrained results.** By problem type and difficulty.

|  | Accuracy | Entropy |
|---|---|---|
| *Problem Difficulty* | | |
| Easy | 0.68 | 1.96 |
| Medium | 0.29 | 3.04 |
| Hard | 0.50 | 2.73 |
| *Problem Type* | | |
| IP | 0.55 | 1.65 |
| LP | 0.42 | 2.17 |
| MIP | 0.52 | 3.32 |

**Sources of error.** To understand where our method fails, we conducted a targeted expert evaluation on 18 autoformulated problems from the ComplexOR benchmark. An optimization expert manually reviewed each model and assessed the correctness of four key components: decision variables, objective function, equality constraints, and inequality constraints. These assessments were also compared against our objective-value-based proxy for correctness.

*Table 3.* **Sources of error in incorrect formulations.**

| Component | Dec var | Obj fun | Eq const | Ineq const | Agree % |
|---|---|---|---|---|---|
| **Error rate** | 23% | 15% | 54% | 54% | 82% |

The expert analysis revealed that constraint modeling, especially inequality constraints, was the most frequent source of error. Issues included incorrect formulations, omissions, or misclassifications (e.g., treating an inequality as an equality), with constraint-related errors present in over 50% of incorrect models. Notably, there was an 82% agreement between the expert's judgments and our objective-value proxy. In two cases, the expert assessed the model to be incorrect despite matching the correct objective value; in two others, models assessed to be correct by the expert produced slightly incorrect objectives. These findings suggest that while accuracy based on comparing returned objective values serve as a strong and scalable proxy for formulation correctness, it does not always capture semantic correctness, highlighting the need for caution when interpreting matching objective values as evidence of fully correct formulations.

## 6. Discussions

In summary, this work formally defines autoformulation for mathematical optimization models, establishing objectives, evaluation metrics, and identifying key challenges. We introduced a novel approach that frames autoformulation as a search problem, effectively leveraging the hierarchical structure of optimization modeling. Our method integrates LLMs as conditional hypothesis generators and evaluators of formulation correctness within an MCTS framework, systematically exploring the hypothesis space of possible formulations. The introduction of search pruning using SMT solvers

further enhances efficiency by eliminating redundant formulations. Empirical evaluations across real-world benchmarks demonstrate our method's superior performance in formulating correct models, with notable efficiency gains from pruning and LLM-based correctness evaluation.

**Future Work.** Looking ahead, we see autoformulation as a promising domain where LLMs can meaningfully augment human expertise. Future research directions include developing collaborative frameworks that integrate humans in-the-loop with autoformulator capabilities, potentially leveraging active acquisition techniques (Astorga et al., 2024; Kobalczyk et al., 2025). Additionally, exploring advanced LLM-based methods, such as retrieval-augmented generation (Lewis et al., 2020), can further enhance formulation processes. Our current method can be viewed as a form of test-time search, and its core principles, hierarchical decomposition, LLM-based evaluation of formulation quality, and search space pruning, can be naturally extended to inform LLM finetuning via outcome- or process-level supervision (Lightman et al., 2023; Wan et al., 2024). Advancing this field will also require the creation of large-scale, diverse benchmarks spanning a wider range of problem types and complexities. In particular, future benchmarks should go beyond integer and mixed-integer programming and include more challenging problems that demand advanced or creative reformulations.

## Impact Statement

While automated optimization model formulation through LLMs offers promising efficiency gains, it raises concerns about model reliability and verification challenges, as LLM-generated formulations may contain subtle errors that could lead to incorrect solutions in critical applications. As formulations grow more complex, verifying their correctness becomes increasingly challenging. Prior to deployment in critical applications, practitioners must have a clear understanding of system capabilities and limitations, alongside robust institutional frameworks that ensure human oversight and expert validation.

## Reproducibility

We provide details on implementing our methods and reproducing results in Section 5 and Appendix B. We provide the code to reproduce our results at https://github.com/jumpynitro/AutoFormulator.[1]

## Acknowledgements

We thank the anonymous ICML reviewers, members of the van der Schaar lab, and Andrew Rashbass for many insightful comments and suggestions. Tennison Liu would like to thank AstraZeneca for their sponsorship and support. Nicolás Astorga thanks W.D. Armstrong Trust for their support. Yuanzhang Xiao was supported by the National Science Foundation under Grant NRT-AI 2244574. This work was supported by Microsoft's Accelerate Foundation Models Academic Research initiative.

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

# A. Additional Results

## A.1. Pass@$N$ *vs.* Best-of-$N$ Results

We conduct two additional analyses: **(1)** evaluating the effectiveness of selecting the best formulation from our method using its estimated reward, and **(2)** comparing our method's performance against ORLM under the Pass@$N$ metric. For fair comparisons, we generate $N$ independent samples from ORLM, matching our rollout count. We focus on ORLM for comparison, since our method already outperforms other baselines at $N = 1$. These comparisons aim to highlight a key distinction: our method performs structured exploration with redundancy pruning, encouraging diversity and functional distinctness among formulations. In contrast, naive sampling (as with ORLM) often produces redundant or similar outputs due to the lack of guided search.

Table 4 reports the Best-of-$N$ results, where we select the top-ranked formulation by estimated score (i.e., $\arg\max$ over the $N$ outputs). Our method selects the best formulation on over $90\%$ of problems, consistently outperforming baselines and supporting the value of our evaluation mechanism. In Table 5, we compare Pass@$N$ performance with ORLM. Our method maintains a clear advantage, further demonstrating its strength in structured, feedback-driven search over functionally diverse solution candidates, an ability ORLM does not inherently possess.

*Table 4.* **Pass@$N$ *vs.* Best-of-$N$.**

| Method | NL4OPT | IndustryOR | MAMO (ComplexLP) | ComplexOR |
|---|---|---|---|---|
| *Pass@$N$ results* | | | | |
| MCTS ($N$=1) | 85.24% | 35.0% | 43.8% | 66.7% |
| MCTS ($N$=3) | 92.21% | 42.0% | 61.4% | 72.2% |
| *Best-of-$N$ results (selected using formulation reward)* | | | | |
| MCTS ($N$=3) | 88.11% | 37.00% | 53.3% | 72.2% |

*Table 5.* **Pass@$N$ comparison to ORLM.**

| Method | NL4OPT | IndustryOR | MAMO (ComplexLP) | ComplexOR |
|---|---|---|---|---|
| ORLM ($N$=1) | 85.7% | 38.0% | 39.3% | 44.4% |
| ORLM ($N$=3) | 90.2% | 42.0% | 56.3% | 61.1% |
| MCTS ($N$=1) | 85.24% | 35.0% | 43.8% | 66.7% |
| MCTS ($N$=3) | 92.21% | 42.0% | 61.4% | 72.2% |

## A.2. Comparisons Against Additional Baselines

In this section of the Appendix, we present additional ablation studies to isolate the key contributors to the performance of our MCTS-based method. We compare against two additional baselines:

1. A Tree-of-Thought (implemented using Depth-First Search) baseline using the same hierarchical structure but without uncertainty guidance or search feedback;

2. A naive sequential sampling baseline with the same hierarchy but no structured search or pruning: each component is sampled sequentially, conditioned only on the partial formulation.

Results in Table 6 show that our method consistently outperforms both baselines across all benchmarks. This highlights the importance of structured decomposition, feedback-driven search, and redundancy pruning. Compared to Tree-of-Thought, MCTS offers more effective exploration by using uncertainty and cumulative feedback to avoid suboptimal branches and refine search paths. The comparison with the naive baseline shows that decomposition alone is insufficient: without guided search or pruning, performance degrades, and manual inspection reveals a higher incidence of invalid or redundant formulations.

*Table 6.* **Benchmark comparisons against additional baselines.**

| Method | NL4OPT | IndustryOR |
|---|---|---|
| *Additional baselines* | | |
| Tree-of-Thought ($N$=3) | 66.53% | 28.0% |
| Sequential ($N$=3) | 60.82% | 28.0% |
| Autoformulator ($N$=1) | 85.24% | 35.0% |
| Autoformulator ($N$=3) | 92.21% | 42.0% |
| Autoformulator (All) | 92.62% | 48.0% |

## B. Additional Details on Method

### B.1. Formulation Equivalence Checks

SMT solvers offer a powerful approach for verifying equivalence between various components of optimization models (Barrett & Tinelli, 2018). These tools can rigorously check if different formulations of objective functions, sets of equality constraints, or sets of inequality constraints are logically equivalent. By encoding the components as logical formulas within appropriate theories (such as linear arithmetic), SMT solvers can determine if the formulations are satisfiable under the same conditions. For objective functions, the solver can check if the difference between two functions is always zero across the feasible region. This is formally described in Equation (5). For constraint sets, it can verify if they define identical feasible regions by checking that each constraint in one set is implied by the other set and vice versa, formally described in Equations (6) and (7). This approach not only ensures the correctness of model transformations or reformulations but also aids in identifying redundant constraints and simplifying complex models. However, the effectiveness of SMT solvers in this context depends on the nature of the optimization problem, as nonlinear or highly complex formulations may pose challenges for current solvers.

1. For objective functions $f^{(i)}$ and $f^{(j)}$:

$$\text{Equivalent}(f^{(i)}, f^{(j)}) \iff \forall \mathbf{x} \in \mathcal{X}, f^{(i)}(\mathbf{x}) = f^{(j)}(\mathbf{x}) \tag{5}$$

2. For sets of equality constraints $g^{(i)} = \{g_k^{(i)}\}_k^K$ and $g^{(j)} = \{g_l^{(j)}\}_l^L$:

$$\text{Equivalent}(g^{(i)}, g^{(j)}) \iff \forall \mathbf{x} \in \mathcal{X}, (\bigwedge_k g_k^{(i)}(\mathbf{x}) = 0) \iff (\bigwedge_l g_l^{(j)}(\mathbf{x}) = 0) \tag{6}$$

3. For sets of inequality constraints $h^{(i)} = \{h_k^{(i)}\}_k^K$ and $h^{(j)} = \{h_l^{(j)}\}_l^L$:

$$\text{Equivalent}(h^{(i)}, h^{(j)}) \iff \forall \mathbf{x} \in \mathcal{X}, (\bigwedge_k h_k^{(i)}(\mathbf{x}) \leq 0) \iff (\bigwedge_l h_l^{(j)}(\mathbf{x}) \leq 0) \tag{7}$$

### B.2. Prompt Design

**Template instruction**

```
I have a problem in operational research:
------
###PROBLEM DESCRIPTION###
------
I have the following formalization:
formalization_dict = {"parameters": {}, "decision_variables2: {}, "objective": {},
"equality_constraints": {}, "inequality_constraints": {}}
```

**Template for generating parameters (depth=0)**

```
You are an optimization modeling expert.  Complete formalization_dict based on
the problem description, you should complete the "parameters" field, which
consists of assigning constants to descriptive variable names.
Only complete "parameters" and nothing else.  Follow these guidelines:
1.  Your primary responsibility is to define all the parameters from the
problem description that will later be used to define decision variables, the
objective, and constraints (both equality and inequality).
2.  You may include additional parameters in a format suitable for facilitating
the subsequent tasks of defining decision variables, the objective function,
and constraints.
3.  For parameters that involve multiple indices (e.g., x[i] or x[i,j]), use
the most appropriate data structure, such as lists, dictionaries, or
dictionaries with tuple keys, to represent them.
```

```
4.  For each parameter, include a clear, descriptive comment explaining its
meaning.
5.  Ensure that the parameter names (keys) are descriptive and intuitive.
Return only the python dictionary update (i.e.,
formalization_dict["parameters"] = ...  )  following the described
requirements.
```

**Template for generating decision variables (depth = 1)**

```
You are an optimization modeling expert.  Complete only the "decision_variables"
field within the "formalization_dict" based on the provided problem
description.
Ensure the decision_variables comprehensively cover all essential elements to
accurately model the optimization problem.
Each key-value pair in the dictionary must adhere to the following structure:

<key>: {
    "description": <description>,
    "type": <type>,
    "iteration_space": <space>
}

The structure should meet these requirements:
1.  Each <key> represents a decision variable that will later be used to
implement the objective, equality, and inequality constraints in a Python
program.
2.  Replace <key> with a symbolic name representing the decision variable.
Ensure that each <key> represents a distinct decision variable with a unique
symbolic name.
3.  Replace <description> with a detailed explanation of the role of the
decision variable in the optimization model.
4.  Replace <type> with a string representing the Gurobi variable type (e.g.,
GRB.INTEGER), as this will be used to create the variable via Gurobi's addVar
function.
5.  If the decision variable is indexed, replace <space> with a string
representing Python for-loop using list comprehension syntax to represent the
index space.  For this, assume direct access to these parameter variables
(i.e., avoid using parameters[variables] syntax).
6.  If the variable is not indexed, set <space> to None.
7.  If the variable is indexed, do not write the index in the symbol (do not
put the index when writing <key>).
8.  You are encouraged to create decision variables that are general.  If two
decision variables represent the same concept write them as one key, creating
an appropriate iteration space.
Return only the Python dictionary update (i.e.,
formalization_dict["decision_variables"] = ...)  following the described
requirements.
```

**Template for generating objective functions (depth = 2)**

```
You are an optimization modeling expert.  Complete only the "objective" field
within the "formalization_dict" based on the provided problem description.
Do not complete any other fields.  Follow these requirements:
1.  Write the objective function mathematically using decision variables.
2.  Preface the key-value pair with a Python comment explaining the rationale
```

behind the objective. DO NOT make a commentary inside the mathematical
description.
3. Use parameter-defined variables instead of hard-coded values. Assume
direct access to these parameter variables (i.e., avoid using
parameters[variables] syntax).
4. The dictionary key must be 'min' or 'max', reflecting the nature of the
objective (minimization or maximization).
5. The dictionary value must be a string representation of the objective
function based on the problem description, written in valid Python syntax.
Return only the Python dictionary update (i.e., formalization_dict["objective"]
= "max": ... or formalization_dict["objective"] = "min": ...) following the
described requirements.

### Template for generating equality constraints (depth = 3)

You are an optimization modeling expert. Complete the formalization_dict by
filling in the equality_constraints field based on the problem description and
the decision variables provided.
These constraints include border constraints, initialization, and equality
constraints derived from the problem description. Do not complete the
"inequality_constraints" field. Follow these requirements:
1. Descriptive constraints: Each key in the dictionary should represent a
unique, clearly named constraint, with the value being a string that describes
the corresponding mathematical equality using "==".
2. Parameter Variables: Use parameter-defined variables instead of hard-coded
values. Assume direct access to these parameter variables (i.e., avoid using
parameters[variables] syntax).
3. Indexed Variables: For indexed decision variables, indicate the index
within brackets (e.g., x[i]).
4. Handling Multiple Constraints: For similar constraints that repeat across
indices or variables, use Python for loops and list comprehensions for
efficient representation.
5. String mathematical description: Note, the value (mathematical
description) should be a single string. DO NOT use .join() or anything else.
Even if it represents multiple constraints using a for loop.
6. No Inequality Constraints: Only define equality constraints. Inequality
constraints will be handled separately by a subsequent expert.
7. Comments: Include a Python comment before each key-value pair, explaining
the rationale behind the constraint.
Return only the Python dictionary update (i.e.,
formalization_dict["equality_constraints"] = ...) following these requirements.
Important: If the problem contains only inequality constraints and no equality
constraints, return: formalization_dict["equality_constraints"] = {None: None}.
This will signal the need to focus on inequality constraints in subsequent
modeling steps.

### Template for generating inequality constraints (depth = 4)

You are an optimization modeling expert. Complete the formalization_dict by
adding the inequality_constraints field based on the problem description.
Follow these requirements:
1. Descriptive constraints: Each key in the dictionary should represent a
unique, clearly named constraint, with the value being a string that describes
the corresponding mathematical inequality.

```
2.  Parameter Variables:  Use parameter-defined variables instead of hard-coded
values.  Assume direct access to these parameter variables (i.e., avoid using
parameters[variables] syntax).
3.  Indexed Variables:  For indexed decision variables, indicate the index
within brackets (e.g., x[i]).
4.  Handling Multiple Constraints:  For similar constraints that repeat across
indices or variables, use Python for loops and list comprehensions for
efficient representation.
5.  String mathematical description:  Note, the value (mathematical
description) should be a single string without using join or anything else.
Even if it represents multiple constraints using a for loop.
6.  Inequality Constraints Only:  Include only inequality constraints.  Exclude
any constraints already covered under equality_constraints.
7.  Comments:  Include a Python comment before each key-value pair, explaining
the rationale behind the constraint.
Return only the Python dictionary update (i.e.,
formalization_dict["inequality_constraints"] = ...)  following these
requirements.
Important:  Think carefully of inequality constraints that are not explicit in
the problem description that should be considered.  If after thinking you
conclude the problem contains only equality constraints and no inequality
constraints, return:  formalization_dict["inequality_constraints"] = {None:
None}.
```

**Template for pruning decision variables**

```
– Objective:
As an expert in optimization modeling, your role is to evaluate multiple sets
of decision variables provided for an operations research problem.  You are
responsible for determining if two or more sets of decision variables should be
grouped together based on their equivalency from an optimization perspective.
– Task Breakdown:
Your grouping decision is critical for assisting a subsequent optimization
expert, who will define the objective function, equality constraints, and
inequality constraints for each group.  To facilitate this process, follow
these precise guidelines:
– Equivalency Criteria:
1.  Same Objectives and Constraints:  Two sets of decision variables should be
grouped together if they result in the definition of the same objective
function, equality constraints, and inequality constraints, even if the
variable names differ.
2.  Conceptual Equivalency:  Variable sets should be grouped together if,
despite having different variable names, they define the same underlying
concepts that ultimately lead to identical objectives and constraints (both
equality and inequality).
3.  Non-Equivalency Conditions:  Two sets of decision variables should not be
grouped together if they lead to differences in any of the following:
Objective function, Equality constraints, Inequality constraints.
4.  Naming Convention Irrelevance:  The names of the decision variables are
irrelevant for grouping purposes.  Only the functional impact of the variables
on the objective function and constraints should be considered.  If two sets of
variables lead to the same results, group them together, even if the names
differ.
```

```
By following these guidelines, you will help ensure that decision variable sets
are clearly classified for the next expert in the process.
Please list your clusters as follows:

###
groups = {
1: group_1,
...,
n: group_n}
###

Where group_i is a python list containing the names (string) of all the set of
decision variables that are equivalent.  One set of decision variables can only
belong to one group.  The list should consider at least one element.
Important:  Think carefully STEP BY STEP about your grouping decision, then
conclude your assessment using the structured format provided above.
Here are the current solutions:

solutions = {}
```

**Template for ranking expanded children nodes**

```
You are an expert in optimization modeling.  Using the formalization_dict as
your current progress, you are tasked with selecting the optimal #VARIABLE#
from the provided options.
Please follow these steps:
1.  Carefully evaluate each potential #VARIABLE#.
2.  Rank the variables from best to worst based on their suitability.
Present your rankings in the following format:

###
rank = {
1: solution_1,
...,
n: solution_n}
###

Where:
- solution_1 represents the best #VARIABLE#.
- solution_n represents the least suitable #VARIABLE#.
 Important:  Think carefully STEP BY STEP about your ranking decision.  Then
conclude by listing the solutions in string format as structured above.
Here are the possible solutions:

solutions = {}
```

### B.3. Benchmarks

- **NL4OPT**: A widely adopted benchmark for Operations Research originating from a NeurIPS competition (Ramamonjison et al., 2023). Since the original NL4OPT provides only mathematical formulations, we utilize the labelled problem set prepared by (Tang et al., 2024). This dataset consists of 289 linear programming problems for which optimal solutions were generated using GPT-4 with the assistance of experts, facilitating evaluation based on execution accuracy.

- **MAMO**: A benchmark specifically designed to assess mathematical modeling capabilities of Large Language Models. It comprises two subsets: 652 easy and 211 complex linear programming problems, each accompanied by optimal solutions. Our experiments focus exclusively on the complex subset due to the easier set is relatively saturated in comparison.

- **IndustryOR**: Introduced in (Tang et al., 2024), this benchmark focuses on industrial applications of Operations Research. It includes 100 real-world problems from 13 distinct industries, primarily covering linear programming (LP), integer

programming (IP), and mixed-integer linear programming (MILP), with the addition of one non-linear programming instance. The dataset categorizes problems into three difficulty levels.

- **ComplexOR**: This dataset encompasses 37 complex Operations Research problems across various application domains, including 12 MILP problems. For our evaluation, we utilize all 18 publicly available problems from this dataset[2].

### B.4. Metrics

- **Execution Accuracy**: The primary evaluation metric, defined by the correctness of executable code generated by a model. A response is deemed correct if its computed optimal value aligns closely with the ground truth solutions provided, allowing a margin of error within 5% following the evaluation protocol used in (Tang et al., 2024) (see official implementation).

- **Pass@k**: This metric assesses inference quality by generating $k$ candidate solutions for each problem. The model is considered successful if at least one of these $k$ candidates achieves correctness based on the execution accuracy criterion.

- **Best of N**: This is an inference strategy where the model generates $N$ candidate solutions. A selection mechanism (which could be another model, a heuristic, or a verification process) then chooses the single "best" solution out of the N candidates. In our case, the selection mechanism computes the average $V$ values along the path from the root node to the final node, which is the complete formulation, selecting the candidate with the highest average.

## C. Categorization of Autoformulation Challenges by Optimization Problem Structure

The exact challenges faced by an autoformulator depends on the nature of the problem $d$. Here, we provide a categorization of optimization problems and their characteristics. To help elucidate different types of problems, we introduce two concepts. First, we define the **set of correct formulations** for a problem $\mathcal{M}(d) \subset \mathcal{M}$ as the set of all equivalent formulations that *correctly* model a problem $d$. Second, we introduce the set of **original forms** $\mathcal{M}_o(d) \subseteq \mathcal{M}(d)$—the set containing the natural representations of the problem, typically the initial models an optimization expert would create. This is a set, as it can contain trivially equivalent formulations. Finally, we partition the set $\mathcal{M}$ into the set of convex problems $\mathcal{M}_{\text{conv}}$ and the set of non-convex problems $\mathcal{M}_{\text{nonc}}$.

1. **Type I problems.** These are problems where the original form is inherently convex, namely $\mathcal{M}_o(d) \subseteq \mathcal{M}_{\text{conv}}$. Examples include certain resource allocation problems that can be naturally formulated as linear programs. The challenge of solving Type I problems is to ensure that the problem is correctly represented (**formulation correctness**, i.e. $\mathcal{H}(d) \cap \mathcal{M}_o(d) \neq \emptyset$), which would entail that it can be efficiently solved to global optimality.
2. **Type II problems.** These are problems where the original form is non-convex, but can be reformulated into an equivalent convex problem, namely $\mathcal{M}_o(d) \subseteq \mathcal{M}_{\text{nonc}}$ but $\mathcal{M}(d) \cap \mathcal{M}_{\text{conv}} \neq \emptyset$. In addition to formulation correctness, another challenge of solving Type II problems is to ensure the autoformulator can identify and apply appropriate reformulation strategies (e.g. change of variables) to transform the non-convex into an *equivalent* convex form, namely $\mathcal{H}(d) \cap (\mathcal{M}(d) \cap \mathcal{M}_{\text{conv}}) \neq \emptyset$. For such problems, evaluation extends beyond correctness to include the ability to achieve **global optimality** through reformulation.
3. **Type III problems.** These are problems where the original form is non-convex and cannot be reformulated into a convex problem, namely $\mathcal{M}(d) \subseteq \mathcal{M}_{\text{nonc}}$. In such cases, there are two general options: a) solve the non-convex problem using general-purpose algorithms (e.g. gradient descent), or b) *relax* into a convex problem that approximates, but is not equivalent to, the original problem (e.g. semidefinite relaxation of a Max-Cut problem (Goemans & Williamson, 1995)).

A crucial nuance here is that mathematically equivalent models, even when both are convex, can exhibit vastly different computational complexities. An example of this is quadratic programming and second-order cone programming (SOCP) reformulations of the same problems (Alizadeh & Goldfarb, 2003). Although mathematically equivalent, SOCP formulations often allow for more efficient solution methods. Therefore, **computational efficiency** is an important evaluation metric across all three problem types, significantly impacting practical utility of model formulations. In Appendix D, we provide concrete examples to illustrate each type of optimization problems.

---

[2]https://github.com/xzymustbexzy/Chain-of-Experts/tree/main

# D. Illustrative Examples of Problem Categorization

In this section, we provide examples of canonical problems in engineering and machine learning that belong to each of the identified problem types. Specifically:

- **Type I**: Problems that have a precise mathematical model, which is convex in its original form. Examples are provided in Appendix D.1.
- **Type II**: Problems that have a precise mathematical model, which is non-convex in its original form but can be reformulated as a convex problem (sometimes additional assumptions are needed). Examples are provided in Appendix D.2.
- **Type III**: Problems that have a precise mathematical model, which is non-convex in its original form but can be *relaxed* to a convex problem (sometimes additional assumptions are needed). The difference from **Type II** is that the convex relaxation is *not* equivalent to the original problem. Examples are provided in Appendix D.3.

## D.1. Examples of Type-I Problems

**Overview.** Maximizing mutual information in a wireless channel.

Mutual information is a quantity that measures the divergence between two random variables, with applications in wireless communications (Goldsmith & Varaiya, 1996) and in data science (Belghazi et al., 2018). Here, we describe it in the context of maximizing Shannon capacity in wireless communications.

We consider a discrete memoryless channel with an input random variable $X \in \{1, \ldots, \ell\}$, an output random variable $Y \in \{1, \ldots, y\}$, and a channel transition matrix $P \in \mathbb{R}^{y \times \ell}$ with the element on the $j$-th row and the $i$-th column being $p_{ji} = \text{prob}\,(Y = j \mid X = i)$.

$$\text{Input } X \longrightarrow \boxed{\text{Transition Probability } P} \longrightarrow \text{Output } Y$$

Our goal is to choose the optimal probability distribution of input $X$, denoted $x \in \mathbb{R}^\ell$ with $x_i = \text{prob}\,(X = i)$, in order to maximize the mutual information between input $X$ and input $Y$

$$I(X; Y) = \sum_{i=1}^{\ell} \sum_{j=1}^{y} x_i p_{ji} \log_2 \frac{p_{ji}}{\sum_{k=1}^{\ell} x_k p_{jk}}.$$

The optimal value of the problem is called Shannon capacity.

This problem is convex in its original form:

$$
\begin{aligned}
\text{Maximize} \quad & \sum_{i=1}^{\ell} \left( \sum_{j=1}^{y} p_{ji} \log_2 p_{ji} \right) x_i - \sum_{j=1}^{y} \left( \sum_{i=1}^{\ell} p_{ji} x_i \right) \log_2 \left( \sum_{i=1}^{\ell} p_{ji} x_i \right) \\
\text{subject to} \quad & x_i \geq 0, \quad i = 1, \ldots, \ell, \\
& \sum_{i=1}^{\ell} x_i = 1.
\end{aligned}
\tag{8}
$$

- **Reformulation strategies:** None.
- **Difficulty in reformulation:** Not applicable.
- **Difficulty in solving the reformulated/original problem:** Easy.

## D.2. Example of Type-II Problem

**Overview.** Power control to satisfy SINR requirements with minimum power usage (`PC-MinPower`)

We consider the problem of determine the transmit power of $\ell$ pairs of transceivers. They operate in the same frequency at the same time, hence causing interference to each other. The problem data is a channel gain matrix $\mathbf{G} \in \mathbb{R}^{\ell \times \ell}$, where $g_{ij}$ is

the channel gain from transmitter $j$ to receiver $i$, the noise power vector $\sigma \in \mathbb{R}^\ell$ with $\sigma_i$ as the noise power at receiver $i$, and the minimum SINR requirement vector $\gamma \in \mathbb{R}^\ell$ with $\gamma_i$ as the minimum SINR required by the transceiver $i$.

Our goal is to choose the transmit power, denoted $x \in \mathbb{R}_+^\ell$ with $x_i$ being the power of transmitter $i$, in order to minimize the total transmit power while satisfying the SINR requirements of each transceiver (Yates, 1995).

This problem is non-convex in its original form:

$$
\begin{aligned}
\text{Minimize} \quad & \sum_{i=1}^{\ell} x_i \\
\text{subject to} \quad & \frac{g_{ii} x_i}{\sum_{j \neq i} g_{ij} x_j + \sigma_i} \geq \gamma_i, \quad i = 1, \ldots, \ell.
\end{aligned}
\tag{9}
$$

But it is not hard to observe that the constraints of SINR requirements can be reformulated as linear constraints, resulting in a LP:

$$
\begin{aligned}
\text{Minimize} \quad & \sum_{i=1}^{\ell} x_i \\
\text{subject to} \quad & g_{ii} x_i \geq \gamma_i \left( \sum_{j \neq i} g_{ij} x_j + \sigma_i \right), \quad i = 1, \ldots, \ell.
\end{aligned}
\tag{10}
$$

- **Reformulation strategies:** Transformation of function.
- **Difficulty in reformulation:** Easy (straightforward observation).
- **Difficulty in solving the reformulated/original problem:** Easy (the reformulated problem is LP).

### D.3. Examples of Type-III Problems

We consider the same setting as `Beamform-MinSidelobe`. But here, our goal is to maximize the gain at the target direction $\theta_{\text{tar}}$, while limiting the ripple effect at directions $\theta_1, \ldots, \theta_m$ outside the target area.

This problem is non-convex: (Fuchs, 2013)

$$
\begin{aligned}
\text{Maximize} \quad & \left| G(\mathbf{w}; \theta^{\text{tar}}) \right| \\
\text{subject to} \quad & 1/\delta \leq |G(\mathbf{w}; \theta_i)| \leq \delta, \ i = 1, \ldots, m.
\end{aligned}
\tag{11}
$$

It is non-convex because we maximize a convex function (i.e., norm) and have constraints on convex functions greater than or equal to a constant.

In this case, the standard semidefinite relaxation technique can be used, which "lifts" the problem to higher dimensions. Specifically, we define a rank-1 semidefinite matrix $\mathbf{W} \triangleq \mathbf{w}\mathbf{w}^H$. Then the gain at direction $\theta$ satisfies

$$
|G(\mathbf{w}; \theta)|^2 = \text{trace}\left(\mathbf{E}(\theta_i) \cdot \mathbf{W}\right),
$$

where $\mathbf{E}(\theta_i) = \mathbf{e}(\theta_i)^* \cdot \mathbf{e}(\theta_i)^T \in \mathbb{C}^{n \times n}$ with

$$
\mathbf{e}(\theta_i) = \left[ e^{\mathbf{i}(x_1 \cos\theta + y_1 \sin\theta)}, \ldots, e^{\mathbf{i}(x_n \cos\theta + y_n \sin\theta)} \right]^T.
$$

With the new matrix variable $\mathbf{W}$, we have the following equivalent problem:

$$
\begin{aligned}
\text{Maximize} \quad & \text{trace}\left(\mathbf{E}(\theta_{\text{tar}}) \cdot \mathbf{W}\right) \\
\text{subject to} \quad & (1/\delta)^2 \leq \text{trace}\left(\mathbf{E}(\theta_i) \cdot \mathbf{W}\right) \leq \delta^2, \ i = 1, \ldots, m, \\
& \mathbf{W} \succeq 0, \\
& \text{rank}(\mathbf{W}) = 1.
\end{aligned}
\tag{12}
$$

Here, the only nonconvexity comes from the rank constraint. By removing it, we get the following convex relaxation:

$$
\begin{aligned}
\text{Maximize} \quad & \text{trace}\left(\mathbf{E}(\theta_{\text{tar}}) \cdot \mathbf{W}\right) \\
\text{subject to} \quad & (1/\delta)^2 \leq \text{trace}\left(\mathbf{E}(\theta_i) \cdot \mathbf{W}\right) \leq \delta^2, \; i = 1, \ldots, m, \\
& \mathbf{W} \succeq 0.
\end{aligned}
\tag{13}
$$

In general, we need to recover an approximate solution vector $\mathbf{w}$ from the solution matrix $\mathbf{W}$. Under certain conditions (e.g., uniform linear arrays), we can guarantee to recover the *exact* solution vector.

- **Relaxation strategies:** Semidefinite relaxation (SDR).
- **Difficulty in reformulation:** Easy (standard SDR techniques were used).
- **Difficulty in solving the reformulated/original problem:** Medium (the relaxed convex problem is a SDP, and recovery methods are needed).

# E. Effect of Formulation on Optimality, Solution Time

In this section, we present simulation results comparing the performance of various optimization solvers on the `PC_MinPower` problem, both in its original non-convex form and in a reformulated convex form (see Appendix D.2). We consider problem instances with $\ell = 10$ and $\ell = 100$ users (i.e., $\ell$ optimization variables). For each instance, we evaluate the solvers in terms of success rate, optimality gap, and average solve time over 100 random samples.

### E.1. Experimental Setup

The `PC_MinPower` problem aims to minimize the total power consumption in a system while satisfying certain constraints. The original formulation of this problem is non-convex, which can pose challenges for optimization algorithms. However, by reformulating the problem, it can be converted into an equivalent convex problem, which is generally easier to solve efficiently.

We evaluated the following solvers:

- **General-Purpose Solvers**:
    - **TRCA**: Trust-Region Constrained Algorithm
    - **SLSQP**: Sequential Least Squares Programming.
    - **COBYLA**: Constrained Optimization BY Linear Approximations.
    - **COBYQA**: Constrained Optimization BY Quadratic Approximations.
- **Convex Program Solvers**:
    - **CLARABEL**: A conic optimization solver.
    - **ECOS**: Embedded Conic Solver.
    - **SCS**: Splitting Conic Solver.
    - **OSQP**: Operator Splitting Quadratic Program Solver.

For each solver and problem instance, we recorded:

- **Success Rate**: The percentage of runs where the solver successfully found a feasible solution.
- **Optimality Gap**: The difference between the objective value obtained by the solver and the known optimal value.
- **Average Solve Time**: The average computation time (in seconds) required by the solver.

### E.2. Results

Tables 7 and 8 present the performance of the solvers for problem sizes $n = 10$ and $n = 100$, respectively.

### E.3. Discussion

The results demonstrate several key observations:

*Table 7.* Solver Performance for $\ell = 10$ Users

| Solver | Type | Original Nonconvex Problem | | | Reformulated Convex Problem | | |
|---|---|---|---|---|---|---|---|
| | | Success | Optimality Gap | Time (s) | Success | Optimality Gap | Time (s) |
| *General-Purpose Solvers* | | | | | | | |
| TRCA | General-Purpose | 100% | $7.31 \times 10^{-3}$ | 0.0399 | 100% | $1.25 \times 10^{-3}$ | 0.0420 |
| SLSQP | General-Purpose | 100% | $6.47 \times 10^{-7}$ | 0.0019 | 100% | $6.48 \times 10^{-7}$ | 0.0009 |
| COBYLA | General-Purpose | 67% | $2.94 \times 10^{-6}$ | 0.0073 | 100% | $7.15 \times 10^{-6}$ | 0.0039 |
| COBYQA | General-Purpose | 0% | — | — | 6% | 9.80 | 14.4067 |
| *Convex Program Solvers* | | | | | | | |
| CLARABEL | Convex Solver | — | — | — | 100% | $6.32 \times 10^{-7}$ | 0.0002 |
| ECOS | Convex Solver | — | — | — | 100% | $6.16 \times 10^{-7}$ | 0.0001 |
| SCS | Convex Solver | — | — | — | 100% | $4.45 \times 10^{-7}$ | 0.0002 |
| OSQP | Convex Solver | — | — | — | 100% | $6.48 \times 10^{-7}$ | 0.0003 |

*Table 8.* Solver Performance for $\ell = 100$ Users

| Solver | Type | Original Nonconvex Problem | | | Reformulated Convex Problem | | |
|---|---|---|---|---|---|---|---|
| | | Success | Optimality Gap | Time (s) | Success | Optimality Gap | Time (s) |
| *General-Purpose Solvers* | | | | | | | |
| TRCA | General-Purpose | 100% | $7.75 \times 10^{-2}$ | 0.6628 | 100% | $1.28 \times 10^{-2}$ | 0.6856 |
| SLSQP | General-Purpose | 100% | $1.04 \times 10^{-6}$ | 0.0750 | 100% | $1.04 \times 10^{-6}$ | 0.0298 |
| COBYLA | General-Purpose | 0% | — | 6.0764 | 100% | $2.86 \times 10^{-5}$ | 9.9629 |
| COBYQA | General-Purpose | 0% | — | — | 0% | — | — |
| *Convex Program Solvers* | | | | | | | |
| CLARABEL | Convex Solver | — | — | — | 100% | $6.22 \times 10^{-7}$ | 0.0121 |
| ECOS | Convex Solver | — | — | — | 100% | $9.91 \times 10^{-7}$ | 0.0097 |
| SCS | Convex Solver | — | — | — | 100% | $1.05 \times 10^{-6}$ | 0.0055 |
| OSQP | Convex Solver | — | — | — | 100% | $1.04 \times 10^{-6}$ | 0.0110 |

- **Importance of Problem Reformulation**: For general-purpose solvers, reformulating the original nonconvex problem into an equivalent convex problem significantly improves solution quality. This improvement is more pronounced for larger problem sizes ($\ell = 100$). For instance, COBYLA's success rate increased from 0% to 100% when the problem was reformulated.

- **Solver Selection Matters**: Different general-purpose solvers exhibit varying performance levels. SLSQP consistently achieves near-zero optimality gaps and high success rates with relatively low solve times across both problem formulations and sizes. In contrast, COBYQA fails to find feasible solutions in most cases, highlighting the necessity of careful solver selection.

- **Performance of Convex Program Solvers**: For the reformulated convex problem, convex program solvers (CLARABEL, ECOS, SCS, OSQP) show excellent and consistent performance. They all achieve 100% success rates, negligible optimality gaps, and minimal solve times. The differences among these solvers are minimal, suggesting that any of them would be suitable for solving the convex formulation efficiently.

These findings underscore the importance of problem reformulation and solver selection in optimization tasks. Reformulating a non-convex problem into a convex one can significantly enhance the performance of general-purpose solvers. Additionally, selecting the appropriate solver is crucial, as it can greatly impact the success rate and computational efficiency.

