# OpenReview forum: "Autoformulation of Mathematical Optimization Models Using LLMs"
_ICML.cc/2025/Conference — ICML 2025 poster_

### Official Review · Reviewer_nBsi · 2025-03-08

**Overall Recommendation:** 2

**Summary:**

This paper proposes an approach for autoformulation—the automated creation of solver-ready mathematical optimization models from natural language problem descriptions. The authors frame autoformulation as a search problem and leverage Large Language Models (LLMs) with Monte-Carlo Tree Search (MCTS) to systematically generate, explore, and evaluate candidate optimization formulations.

**Claims And Evidence:**

- I personally disagree with the claim that "optimization modeling follows a three-step process." Recent papers in operations research primarily focus on proving why their (human-derived) formulations are theoretically correct. While understanding the problem description in text and generating execution code can be useful, the core aspect of optimization modeling—the theoretical justification of the formulation—is entirely missing from this perspective.

- A key limitation of this work is that the task itself may not reflect a real-world need—one that domain scientists would actually use. Can you justify the practical use case of such a system and provide an estimate of who, and how many, would realistically adopt it? Otherwise, creating such a task solely for benchmarking against potentially irrelevant metrics raises concerns about its practical impact.

**Essential References Not Discussed:**

NA

**Experimental Designs Or Analyses:**

The authors empirically evaluate the proposed method and demonstrate that it achieves better results.

**Methods And Evaluation Criteria:**

yes.

**Other Comments Or Suggestions:**

NA

**Other Strengths And Weaknesses:**

**Strengths:**
- The experimental evaluation is solid.

**Weaknesses:**
- The paper is purely empirically driven. Given its focus, it may be more suitable for an EMNLP conference.

**Questions For Authors:**

There is already extensive research on using LLMs with MCTS across various tasks aimed at improving automation, so the main algorithm in this work is not novel.

Additionally, a key limitation of this work is that the task itself may not be a *real* task—one that domain scientists would actually use. Why create such a task solely for benchmarking against metrics that may lack practical relevance?

**Relation To Broader Scientific Literature:**

The developed software can be beneficial to the operations research community.

**Theoretical Claims:**

No theory is provided in this paper.

---

> ### Author Rebuttal · Authors · 2025-03-31
>
> *We appreciate the reviewer’s detailed and thoughtful evaluation.*
>
> ---
>
> ### [P1] Engineering vs theory
>
> Thank you for the thoughtful comment. Our framing of optimization modeling as a three-step process (requirements gathering $\rightarrow$ mathematical model $\rightarrow$ computational model) is intended to reflect the **engineering** side of the modeling pipeline. We agree that in many academic OR contexts, the emphasis lies in **theoretical** justifications—such as proving optimality bounds, analyzing relaxations, or exploring structural properties—which are essential to advancing the science of optimization.
>
> While theoretical work is crucial, we also believe our focus remains valuable (further discussed in **[P2]**). For domain scientists who are not optimization experts, this technology can help democratize modeling by lowering the barrier to entry. For experts, the autoformulator can accelerate prototyping and iteration. When theoretical validation is ultimately required, our system can automate time-consuming or error-prone steps, surface promising candidate formulations, and free up modeling experts to focus on deeper theoretical analysis.
>
> **Actions taken.** We have updated the manuscript to acknowledge this theoretical dimension more clearly. Specifically, we revised lines `L10–L16 R` (Introduction) to highlight the role of theoretical modeling and added a discussion in `L405–L420 R` (Discussion) on future directions that connect autoformulation with theoretical reasoning.
>
> ---
>
> ### [P2] Real-world utility
>
> **Practical utility.** We believe that in many domains—such as logistics, healthcare, energy, and manufacturing—there exists a real and growing need to bridge the gap between domain expertise and optimization modeling expertise. Gurobi’s "State of Optimization Report 2023" [R1–R2], based on a survey of 394 commercial users, identified optimization modeling as a rapidly growing area of demand, while also highlighting a significant skills gap. Just as coding LLMs have empowered non-expert programmers to build software, autoformulators can lower the barrier to modeling, reduce translation bottlenecks, and empower domain experts to experiment with formal optimization tasks. For experienced modelers, these systems can accelerate iteration, support exploratory modeling, and free cognitive effort for higher-level design and analysis.
>
> **Emerging research & industry focus.** Autoformulation is gaining traction in both research and industry. On the research side, recent systems such as Optimus and Chain-of-Experts (both appeared at ICLR 2024) highlight growing interest in the scientific and technical challenges of this space. We note that the benchmarks, tasks, and metrics we used are consistent with those employed in prior works. Industry interest further reinforces the relevance of this problem: (1) Gurobi’s modeling assistant [R3], (2) AWS’s optimization AI tools [R4], and (3) IBM’s research on AI-assisted modeling [R5] all address related aspects of the autoformulation pipeline. We view these developments as strong signals of real-world utility, and our system contributes foundational capabilities to this emerging and impactful area.
>
> **Actions taken.** Thank you for raising this important point. We have added a new appendix section outlining practical and research use cases of our system.
>
> [R1, R2] stage.gurobi.com/resources/report-state-of-mathematical-optimization-2023/, stage.gurobi.com/resources/report-state-of-mathematical-optimization-in-data-science-2023/
>
> [R3] gurobi-ai-modeling.readthedocs.io
>
> [R4] aws.amazon.com/cn/blogs/industries/optimization-in-the-era-of-generative-ai/
>
> [R5] research.ibm.com/publications/enhancing-decision-making-through-the-integration-of-large-language-models-and-operations-research-optimization-bridge-talk
>
> ---
>
> ### [P3] Highlighting novelty
>
> We introduced a novel MCTS-based technique tailored to the structure of optimization modeling tasks, grounded in the core challenges **[C1-C3]**, including (1) domain-specific MCTS strategy to enhance hierarchical exploration; (2) SMT-based pruning of trivial equivalences, improving search efficiency by two orders of magnitude; and (3) comparative scoring of candidate formulations, enhancing feedback for search guidance. Empirically, we show that generic, off-the-shelf approaches fail to address key challenges in evaluating formulation quality (`S5.2`) and eliminating trivial redundancies (`Fig. 4`). In contrast, our approach outperforms prior state-of-the-art methods—including LLMs specifically finetuned for autoformulation—achieving new best results on formulation correctness across multiple challenging benchmarks.
>
>
> ---
>
> *Thanks again; we hope our responses addressed your concerns and would appreciate your consideration in updating the score. We welcome further discussions.*

---

> > ### Comment · Reviewer_nBsi · 2025-04-02
> >
> > Thanks for answering my questions. As you mentioned in your reply—referring to Gurobi and the Amazon link—this work might be better appreciated by a conference in operations research, where the problem description itself may evolve over time, making classic formalizations less effective. In such cases, a tool for automatic problem formalization is essential, aligning with the field's emphasis on future directions in automation and the auto-formation of mathematical optimization problems.
> >
> > Regarding the technical innovation, I personally feel that the contributions are primarily focused on automating the optimization and problem-formalization pipeline. As a result, it may be challenging for this community to extract new insights from the work, making it feel relatively less novel.

---

> > > ### Author Response · Authors · 2025-04-03
> > >
> > > Thank you for your thoughtful follow-up. We appreciate your acknowledgment that the autoformulator aligns with future-facing research directions, particularly in automating mathematical problem formalization.
> > >
> > > ---
> > >
> > > ### [P4] Suitability for conference
> > >
> > > We first want to highlight that our submission is aligned with the conference's [Call for Papers](https://icml.cc/Conferences/2025/CallForPapers), which explicitly welcomes `Application-Driven Machine Learning`, defined as “innovative techniques, problems, and datasets that are of interest to the machine learning community and driven by the needs of end-users.” Our work, under the “Applications” primary area, directly responds to this call by introducing novel methods to address the growing gap between domain expertise and optimization expertise in fields like logistics, healthcare, and engineering.
> > >
> > > This line of work has also gained momentum at recent ML venues, which further supports its suitability to be presented at ICML 2025:
> > > * **NL4OPT** (NeurIPS 2022, Competition Track)
> > > * **Chain-of-Experts** (ICLR 2024)
> > > * **Optimus** (ICML 2024)
> > > * **LLMOPT** (ICLR 2025)
> > >
> > > These papers explore natural language to optimization model pipelines using LLM frameworks. Our method extends this thread by introducing hierarchical search, comparative scoring, and symbolic pruning, enabling tractable and scalable exploration of the formulation space.
> > >
> > > ---
> > >
> > > ### [P5] Technical novelty
> > >
> > > While our method builds on known components, its primary ML contribution lies in combining neural and symbolic reasoning in a domain-specific structure, where generic approaches fail. This is evidenced both empirically (e.g., comparative results in `S5.2` and `Fig. 4`) and technically (e.g., SMT-based pruning, comparative scoring, and domain-specific MCTS), culminating in state-of-the-art performance across multiple competitive benchmarks.
> > >
> > > More broadly, our work is relevant to an important ML research trend: building compound systems capable of structured reasoning in real-world domains. In areas like program synthesis, tool-using agents, and scientific modeling, research increasingly focuses on how to scaffold LLMs with **symbolic tools** and **problem-specific structure**—from execution traces and solver feedback to formal mathematical constraints—to produce outputs that are not only fluent, but also correct, aligned, and executable. As such, beyond the autoformulation-specific utility, we believe our work also offers generalizable insights for structured reasoning tasks—through exploiting domain-specific structure, symbolic tools, and efficient search algorithms.
> > >
> > > ---
> > >
> > > We believe this community is well-positioned to advance interdisciplinary research that intersects learning, structured search, and symbolic reasoning—and we would be eager to contribute to that conversation.

---

### Official Review · Reviewer_14k8 · 2025-03-10

**Overall Recommendation:** 2

**Summary:**

This paper proposes a search-based autoformulation of mathematical optimization problems. The authors provide a formal definition of autoformulation and use MCTS to construct the formulation. Experiments demonstrate the method can outperform the baselines.

**Claims And Evidence:**

The paper proposes three challenges of autoformulation. "(1) the vast, problem-dependent hypothesis space, (2) efficient and diverse exploration of this space under uncertainty, and (3) evaluation of formulation correctness against problem description." However, the paper seems to address only the second challenge.

**Essential References Not Discussed:**

None

**Experimental Designs Or Analyses:**

1. The dataset is limited. The paper only conducts experiments on two datasets in the main paper and more datasets are needed, such as the MAMO dataset and ComplexOR dataset.
2. Some experimental results are missing. I wonder whether the results of Reflextion/Chain-of-Experts/Optimus are missing on the IndustryOR dataset.

**Methods And Evaluation Criteria:**

1. My biggest concern is the unfair experiment settings. The MCTS can have many rollouts. If I understand correctly, the authors think it is successful in solving the problem if any one of the rollouts is correct. This is unfair since the baselines have only one "rollout" during evaluation. As shown in Figure 5, if we set the rollout number to be 1, the accuracy in Figure 5 is under 36%, which is inferior to the 38% of ORLM.
2. The author may want to clarify how to choose the best answer in the rollouts. In my understanding, the method should provide a final answer among the answers given in different rollouts, instead of providing all the answers as final outputs.
3. In this paper, the depth of the MCTS is under 5. The complex MCTS seems not necessary for the shallow search depths.

**Other Comments Or Suggestions:**

Please see the comments above. I suggest the authors refine the experiments part of this paper.

**Other Strengths And Weaknesses:**

None

**Questions For Authors:**

Please see the comments above.

**Relation To Broader Scientific Literature:**

This paper is related to autoformulation for operations research problems.

**Theoretical Claims:**

This paper does not contain any proof for theoretical claims.

---

> ### Author Rebuttal · Authors · 2025-03-31
>
> *We appreciate the reviewer’s detailed and thoughtful evaluation and positive feedback.*
>
> ---
>
> ### [P1] Addressing challenges
>
> Thank you for raising this point. However, we believe this may stem from a misunderstanding. Please allow us to clarify how our method addresses all three challenges:
>
> | Challenge | Description | Our approach | Location in paper |
> |----------|-------------|--------------|-------------------|
> | **[C1] Problem-dependent hypothesis space** | The formulation space is vast and problem-specific, making manual definition infeasible. | LLMs serve as problem-conditioned hypothesis generators that implicitly define the hypothesis space; candidate components are expanded hierarchically.| `L238–L259 L` |
> | **[C2] Efficient search** | Explore the space efficiently under uncertainty, avoiding redundant or unpromising paths. | MCTS over a hierarchical decomposition balances exploration-exploitation; symbolic pruning eliminates trivial equivalences. | `L182–L208 R`; `L260 L–L240 R` |
> | **[C3] Model evaluation** | Evaluate whether generated formulations faithfully represent the original problem. | Partial evaluation via node-level ranking during expansion; comparative scoring of complete formulations. | `S3.2.2`; `S3.2.3` |
>
> ---
>
> ### [P2] Evaluations
>
> Thank you for pointing this out—we agree with your concern. Our method was originally evaluated using a pass@N metric (i.e., success if any of N iterations is correct), while baseline comparisons were reported under pass@1.
>
> **Actions taken.** We have re-evaluated ORLM under pass@N by generating N independent samples to match the iteration count (Note: our method still outperforms non-ORLM baselines at N=1). We report in [this table](https://imgur.com/a/1DUogjH) this fairer comparison (now added to the appendix), and clarified evaluation metrics in the main text. We observe that our approach maintains a clear advantage even under pass@N. This reflects a core strength of our approach: it performs structured exploration over functionally diverse formulations, which ORLM does not inherently support. We appreciate your feedback, it helped improve the rigor of our evaluation.
>
> ---
>
> ### [P3] Selection
>
> In our method, each MCTS iteration produces a complete formulation along with a score from our evaluation mechanism, allowing us to select the highest-scoring one if desired. **Actions taken.** We now report best-of-N accuracy based on score-ranked outputs in [this table](https://imgur.com/a/0haw2V6) (i.e., selecting best-of-N formulations using greedy selection). Our selected formulations continue to outperform baselines (specifically, our mechanism selects the best formulation on >90% of problems), supporting the effectiveness of our scoring mechanism.
>
> ---
>
> ### [P4] MCTS
>
> Thank you for raising this point. We refer the reviewer to our response to Reviewer `fWQR` (**[P1]**) for a detailed discussion of why optimization modeling benefits from hierarchical decomposition and MCTS-based structured exploration. **Empirical evidence.** We include two additional baselines: (1) a Tree-of-Thought using a different tree search strategy, and (2) a naive sequential sampling baseline without tree search. Our method consistently outperforms both, indicating that even at shallow depths, our MCTS framework yields meaningful gains through principled exploration and pruning.
>
> ---
>
> ### [P5] Additional benchmarks
>
> We agree that broader benchmarking is important. In response, we added experiments on both MAMO (focusing on the more challenging ComplexLP subset) and ComplexOR. Due to time constraints, we prioritized ComplexLP over EasyLP, which is relatively saturated. On both benchmarks, our method outperforms all baselines, further validating its effectiveness. Results have been added to the updated [Table 1](https://imgur.com/a/Rl5AGhv).
>
> ---
>
> ### [P6] Other comments
>
> * **Baseline results on IndustryOR.** We attempted to run Chain-of-Experts and Optimus on IndustryOR using their released code but encountered compatibility issues. As noted by the IndustryOR authors, these methods require non-trivial adaptation. Due to time constraints, we were unable to complete this during the rebuttal, but we plan to include these results in a future version (if accepted).
> * **App D.** App D is not meant to evaluate our method directly, but to explore how solver and formulation choices impact downstream performance. While not central, we included it for context and will trim it in a future version to improve clarity.
>
> ---
>
>
> *Thanks again; we hope our responses addressed your concerns and would appreciate your consideration in updating the score. We welcome further discussions.*

---

### Official Review · Reviewer_Pqeu · 2025-03-17

**Overall Recommendation:** 4

**Summary:**

This work studies the autoformulation for mathematical optimization models, or the task of building an optimization model from natural language prompts describing the problem. The approach begins by defining the construction of an optimization model as a hierarchical task, with steps of selecting parameters and decision variables, an objective function, equality constraints, and inequality constraints. These tasks are treated hierarchically using Monte Carlo tree search, where LLMs are used to generate candidate nodes. The authors provide some creative ways to prune trivially equivalent nodes, as well as to score the candidate nodes.

**Claims And Evidence:**

The claims that the tree search, pruning steps, and scoring steps are all effective are well evidenced by comparison against LLM benchmarks and ablation studies in Sections 5.1-5.4.

**Essential References Not Discussed:**

NA

**Experimental Designs Or Analyses:**

The experiments are designed well, and several ablation experiments are performed. It would be interesting to see more evaluation criteria in addition to just having the correct objective value. For example, how many of the models correctly model, overapproximate, or underapproximate the feasible region? How many of the models have variables and parameters correctly defined?

**Methods And Evaluation Criteria:**

The proposed method is tested on two problem sets, NLP4OPT and IndustryOR, and compared against several recent methods for autoformulation. The main evaluation criterion used is the proportion of generated formulations returning the correct optimal objective value. While this is an easy proxy for model correctness, it would be more interesting to have even a limited number of instance that are checked by a human optimization expert (i.e., is this the model an expert would have built?). Moreover, the majority of models tested are linear programs; it would be more interesting to see how the proposed approach generalizes (or requires improvements) in nonlinear or discrete settings.

**Other Comments Or Suggestions:**

NA

**Other Strengths And Weaknesses:**

NA

**Questions For Authors:**

NA

**Relation To Broader Scientific Literature:**

This paper follows on a recent line of work on autoformulation of optimization models.

**Theoretical Claims:**

The only theoretical claim made appears to be that the proposed SMT-solver can correctly prune trivially equivalent models, which is described in the Appendix.

---

> ### Author Rebuttal · Authors · 2025-03-31
>
> *We appreciate the reviewer’s detailed and thoughtful evaluation and positive feedback.*
>
> ---
>
> ### [P1] Additional analysis
>
> Thank you for this thoughtful suggestion. While objective-value correctness is a standard metric in prior work, we agree that it is an imperfect proxy. To address this, we conducted a targeted expert evaluation on 18 autoformulated problems from the ComplexOR benchmark.
>
> An optimization expert manually reviewed each generated model and assessed the correctness of its major components—**(1)** decision variables, **(2)** objective function, **(3)** equality constraints, and **(4)** inequality constraints—along with an overall correctness label. This allowed us to identify sources of modeling errors and compare agreements between expert assessments and our objective-value-based proxy.
>
>
> | Component | Dec var | Obj fun | Eq const | Ineq const | Agreement % |
> |---|---|---|---|---|---|
> | Error rate among incorrect models | 23% | 15% | 54% | 54% | 82% |
>
> **Analysis.** The expert's analysis indicates that the most common sources of error were related to constraint modeling---particularly inequality constraints (e.g., misclassification as equality constraints, incorrect formulation, or omission)---with both equality/inequality constraint errors accounting for over half of all incorrect formulations. Additionally, we observed 82% agreement between the expert’s correctness judgments and the objective-value proxy. In two cases, the expert flagged structural errors despite a correct objective value; in two others, the expert deemed the model correct despite a mismatch in optimal objective values. This suggests that objective-value correctness is a useful but imperfect proxy for full structural accuracy.
>
> **Actions taken.** We have incorporated these findings into the manuscript and now highlight expert evaluation as a valuable future direction for benchmark development and model assessment.
>
> ---
>
> ### [P2] Other problem types
>
> Thank you for this observation. Existing benchmarks in this space—NLP4OPT, IndustryOR, MAMO, and ComplexOR—currently consist exclusively of LP and MILP problems, which shaped the scope of our evaluation. Expanding to nonlinear or more complex problems is an important direction for future work, though it first requires the development of suitable benchmarks.
>
> We note that LPs and MILPs remain highly relevant in practice: according to [R1], 61% of real-world OR problems are MILPs and 41% are LPs. These classes also present the same meaningful modeling challenges **[C1-C3]** for autoformulation, making them a useful starting point.
>
> **Additional results.** That said, we agree that broader evaluations are valuable. Based on your comment and related feedback, we extended our experiments to additional datasets to further strengthen our evaluations. While these new benchmarks, ComplexOR and the ComplexLP subset of MAMO, still consist of LPs and MILPs, they feature more complex problems and serve as a stronger assessment of generality. Our method outperforms all baselines on both, as shown in our updated [Table 1](https://imgur.com/a/Rl5AGhv).
>
> [R1] Gurobi State of Mathematical Optimization 2023 Report https://stage.gurobi.com/resources/report-state-of-mathematical-optimization-2023/
>
> ---
>
> *Thanks again; we hope the reviewer’s concerns are addressed and welcome further discussions.*

---

### Official Review · Reviewer_fWQR · 2025-03-20

**Overall Recommendation:** 3

**Summary:**

This paper introduces autoformulation, the automated translation of natural language problem descriptions into solver-ready mathematical optimization models, addressing the reliance on human expertise in traditional modeling. The proposed method integrates LLMs with MCTS to hierarchically decompose and systematically explore the model space, enhanced by symbolic pruning to eliminate redundant formulations and LLM-based evaluation for guided search. Empirical results on linear and mixed-integer programming benchmarks demonstrate significant performance improvements over existing approaches, showcasing the effectiveness of combining hierarchical exploration, pruning, and dual reward mechanisms.

**Claims And Evidence:**

see the section Strengths And Weaknesses

**Essential References Not Discussed:**

Essential References are well-discussed.

**Experimental Designs Or Analyses:**

Reasonable experimental designs and analyses

**Methods And Evaluation Criteria:**

Yes

**Other Comments Or Suggestions:**

No.

**Other Strengths And Weaknesses:**

### **Strengths**

1. The paper is well-organized and easy to follow, with a clear structure that enhances readability.
2. It gives a deep analysis of the current challenges in this direction
3. Comprehensive ablation experiments are conducted to evaluate each component of the proposed pipeline.

### **Weaknesses**

I am not an expert in this field, so my current stance is a tentative weak acceptance. My recommendation could change following insights from other reviewers’ comments or if the authors provide more clarifications during the rebuttal period.

The authors propose a pipeline that leverages LLMs integrated within an MCTS framework for autoformulation tasks. However, I have several concerns that merit further explanation. First, the tree search employed in the proposed method is notably shallow (consisting of only four layers). I am curious about how the use of MCTS in this context yields substantial benefits over more straightforward or naive search algorithms.

Furthermore, the pipeline bears some resemblance to the tree-of-thought framework, albeit with what appears to be more advanced search strategies tailored specifically for autoformulation. I would appreciate a detailed comparison between the proposed method and the tree-of-thought approach. Including the tree-of-thought method as one of the baseline comparisons in the experimental evaluation would provide a more comprehensive context for assessing the merits of the proposed approach.

Additionally, I suggest that the authors enhance the figure captions with more detailed explanations. As it stands, I found myself repeatedly referring back to the main text to fully understand the figures.

**Questions For Authors:**

N/A

**Relation To Broader Scientific Literature:**

This paper bridges advancements in LLMs and MCTS by introducing a hierarchical, pruning-enhanced framework for automated optimization modeling, extending their applications to mathematical formulation challenges while addressing efficiency and correctness evaluation gaps in prior autoformulation research.

**Theoretical Claims:**

N/A

---

> ### Author Rebuttal · Authors · 2025-03-31
>
> *We appreciate the reviewer’s detailed and thoughtful evaluation and positive feedback.*
>
> ---
>
> ### [P1] MCTS motivation
>
> Thank you for the thoughtful question. We interpret this as raising two related concerns: **(1)** why use a tree-based search framework at all, and **(2)** why choose MCTS specifically.
>
> **(1) Tree-based search rationale.** Although the search depth is modest, optimization models naturally decompose into hierarchical components—decision variables, objective function, and constraints—making tree-based search well-suited. This structure enables:
> * Clearer credit assignment, isolating which components contribute to success;
> * Efficient feedback sharing across subtrees;
> * Tractable redundancy pruning via SMT, which operates more effectively at the component level.
>
> These factors contribute to focused exploration, efficient reuse of components, and better search guidance. These benefits are lost when searching directly over complete formulations, resulting in a flat, entangled space.
>
> **(2) Why MCTS.** We choose MCTS over alternative tree search methods (e.g., DFS, BFS, A*) for two reasons:
> * Exploration under uncertainty. MCTS balances exploration and exploitation using value estimates and UCB, making it better-suited for the inherent ambiguity in autoformulation, as it progressively focuses on promising branches. This stands in contrast to deterministic tree-based algorithms.
> * Feedback-driven search. MCTS uses backpropagation to update its search policy, whereas other methods follow fixed traversal policies (without adjusting the search from observed outcomes).
>
> **Empirical evidence.** We point to three existing lines of evidence supporting the efficacy of our MCTS framework:
> * Sustained exploration: our method continues discovering novel, functionally distinct formulations over iterations (`Fig 5`);
> * Search efficiency: SMT-based pruning reduces redundancy by two orders of magnitude, improving efficiency (`Fig 4`);
> * Improved search guidance: our comparative scoring method provides more robust feedback, enhancing search guidance (`S5.2`).
>
> **Additional results.** Following your feedback, we further isolate our MCTS method's contributions through two baselines:
> * A Tree-of-Thought (DFS) baseline with the same hierarchy but no uncertainty guidance or search feedback;
> * A naive sequential sampling baseline (same hierarchy) without structured search or pruning (i.e., each component is sampled sequentially, conditioned on the partial formulation).
>
> Results from these additional comparisons are provided in **[P2]** and are referenced for further analysis.
>
>
> ---
>
> ### [P2] Tree-of-Thought
>
> Thank you for this comment. Please find a conceptual comparison below:
>
> | **Dimension** | **Autoformulator** | **Tree-of-Thought (ToT)** |
> |---|---|---|
> | **Tree structure** | Structured, domain-specific: variables → objective → constraints | Free-form, unstructured intermediate thoughts |
> | **Node expansion** | LLM-generated components + SMT-based pruning | LLM-generated thoughts; no symbolic pruning |
> | **Node evaluation** | LLM-based comparative scoring of nodes and full models | LLM self-evaluation of nodes in isolation |
> | **Search strategy** | MCTS with UCT (uncertainty-guided, feedback-driven) | Greedy, DFS, or BFS; no feedback or uncertainty modeling |
>
> Compared to ToT, our approach incorporates (1) structured, domain-specific decomposition, (2) uncertainty-aware, feedback-guided exploration, and (3) symbolic pruning to improve efficiency and reduce redundancy.
>
> **Empirical comparison.**  We include both Tree-of-Thought and the sequential sampling baseline in our [updated results](https://imgur.com/a/2NjUrAA). Our method consistently outperforms both across all benchmarks, underscoring the importance of structured decomposition, feedback-driven search, and redundancy pruning. Compared to ToT, MCTS enables more effective exploration by leveraging uncertainty and cumulative feedback to avoid suboptimal branches and refine search trajectories. The comparison with the naive baseline highlights the limits of decomposition alone: without guided search or pruning, performance deteriorates, and manual inspection reveals more invalid or redundant formulations.
>
> **Actions taken.** We have integrated these additional baseline results into the updated manuscript in `Table 1`.
>
> ---
>
> ### [P3] Other comments
>
> * **Figure captions:** Thank you for this suggestion; we have revised the captions to include more detailed and self-contained explanations.
>
> ---
>
> *Thanks again; we hope our responses addressed your concerns and would appreciate your consideration in updating the score. We welcome further discussions.*

---

> > ### Comment · Reviewer_fWQR · 2025-04-02
> >
> > Thank you for the detailed responses. With my concerns addressed, I have decided to maintain my current score.

---

> > > ### Author Response · Authors · 2025-04-04
> > >
> > > Thank you, we are glad we could address your concerns and appreciate your constructive feedback that made our work better!
> > >
> > > The Authors of #10013

---

### Decision · Program_Chairs · 2025-05-01

**Decision:**

Accept (poster)

**Comment:**

**(a) Summary**
This paper introduces an approach for the autoformulation of mathematical optimization problems: translating natural language problem descriptions into solver-ready optimization models.
It proposes a novel framework combining Large Language Models (LLMs) with Monte-Carlo Tree Search (MCTS), guided by symbolic pruning and comparative scoring. The method addresses the key challenges in autoformulation: vast hypothesis space, efficient search under uncertainty, and evaluation of correctness. Empirical results across multiple benchmarks (e.g., NLP4OPT, IndustryOR, MAMO, ComplexOR) demonstrate state-of-the-art performance, and the paper further validates results through ablations and expert evaluations.

**(b) Strengths**
* Clear and well-structured presentation (Reviewer fWQR): The paper is easy to follow and offers an in-depth discussion of core challenges in autoformulation.
* Strong empirical results (Reviewers Pqeu, fWQR, nBsi): The method consistently outperforms baselines across standard and complex benchmarks.
* Methodological novelty (Reviewers fWQR, Pqeu): The use of hierarchical MCTS, symbolic pruning, and comparative scoring is well-motivated and effective.
* Broader relevance (Reviewer nBsi): The paper contributes to the growing interest in combining LLMs with symbolic reasoning for automation in domain-specific tasks.

**(c) Weaknesses**
* Concerns on fairness in evaluation (Reviewer 14k8): Initially, the comparison with baselines was unfair due to differing rollout strategies. The authors addressed this by harmonizing evaluation metrics (pass@N) and adding best-of-N selection.
* Limited scope of datasets (Reviewer 14k8): The original version focused mainly on LPs and MILPs. This was addressed with new results on ComplexLP and ComplexOR.
* Lack of theoretical foundation (Reviewers nBsi, 14k8): The paper is largely empirical and lacks formal theoretical contributions.
* Practical utility questioned (Reviewer nBsi): One reviewer raised concerns about the real-world demand for such autoformulation systems. The authors responded with industrial surveys (e.g., Gurobi 2023) and comparisons to similar efforts at AWS and IBM to support relevance.
* Redundancy of complex search (Reviewer fWQR, 14k8): Concerns were raised about the necessity of MCTS given shallow search depth. The authors defended the choice through empirical comparisons and structured design benefits.

**(d) Decision**
Overall, the reviewers found the paper to be a strong empirical contribution with a clear methodology and promising results. The integration of LLMs, symbolic pruning, and search strategies is well executed and demonstrates performance improvements across diverse benchmarks. Despite lacking theoretical depth and some early concerns about fairness in evaluation and generalizability, the authors have responded thoroughly during the rebuttal period. Given its novel framing, practical impact, and strong empirical support, I recommend acceptance.

**Additional Comments on Reviewer Discussion:**
* fWQR: Appreciated the authors’ clarifications and chose to maintain their weak accept (score 3) after rebuttal.
* Pqeu: Acknowledged the expert evaluation and extended benchmarks, maintaining a score of 4 (accept).
* 14k8: Despite detailed responses and additional experiments by the authors, did not update their weak reject (score 2).
* nBsi: After a nuanced discussion, agreed the work may be better suited to OR venues but acknowledged the application potential. Maintained weak reject (score 2).

Given the split in reviewer scores (4, 3, 2, 2) but a consistently strong author rebuttal and the innovative nature of the work, this paper represents a valuable contribution to ML-driven automation and deserves a place at the conference.